# Melatonin-Mediated Modulation of Grapevine Resistance Physiology, Endogenous Hormonal Dynamics, and Fruit Quality Under Varying Irrigation Amounts

**DOI:** 10.3390/ijms252313081

**Published:** 2024-12-05

**Authors:** Yajuan Chen, Jiangshan Yang, Zhang Shao, Zibo Dai, Dou Li

**Affiliations:** College of Horticulture, Gansu Agricultural University, Lanzhou 730070, China; 107332202377@st.gsau.edu.cn (Y.C.); 1073323120886@st.gsau.edu.cn (Z.S.); 107332212827@st.gsau.edu.cn (Z.D.); 107332102351@st.gsau.edu.cn (D.L.)

**Keywords:** melatonin, irrigation amount, reactive oxygen species, antioxidant enzymes, gene expression

## Abstract

Grapevine, as a globally significant economic fruit tree, is highly sensitive to water stress, which not only damages its growth but also affects the formation of fruit quality. Melatonin (MT), acting as a signaling molecule, plays a crucial role in plant stress responses. However, the regulatory mechanisms of MT on grape leaf physiological characteristics and fruit quality under different irrigation amounts have not been fully elucidated. In this experiment, grape leaves were treated with a 150 μmol·L^−1^ MT solution at 0, 30, 60, and 90 days after flowering under different irrigation amounts (360, 300, 240, and 180 mm). It was found that MT significantly increased the contents of osmotic adjustment substances in leaves, reduced the level of reactive oxygen species, enhanced the activity of antioxidant enzymes, and promoted the metabolism of the ascorbic acid–glutathione cycle, thereby improving the antioxidant capacity of grapes and effectively alleviating the damage caused by a water deficit. At the same time, MT also maintains the dynamic balance of endogenous hormones by upregulating and downregulating the expression levels of related genes, thereby improving fruit quality. In summary, this study reveals the potential application value of MT in enhancing the drought resistance and fruit quality of grapes.

## 1. Introduction

Water is essential for plant survival and a predominant factor in limiting agricultural production [1]. Grapevine is an economically significant fruit tree with a high water requirement. The increasing frequency of extreme weather events due to global warming, coupled with the widespread cultivation of grapevines in arid and semi-arid regions, undoubtedly poses a significant threat to their growth and development. Specifically, the arid climate, the significant imbalance between agricultural and ecological water requirements, and the low annual precipitation in Gansu Province’s Hexi Corridor continue to be the primary environmental challenges affecting grapevine growth. A water deficit can lead to a reduction in plant photosynthetic capacity, disruption of physiological metabolism, damage to the antioxidant system, and imbalances in endogenous hormone levels, consequently hastening leaf senescence [2,3,4]. Conversely, excessive irrigation can result in root hypoxia, root rot, and a decrease in fruit quality, as well as the wastage of water resources and loss of nutrients [5]. In recent years, numerous studies have investigated the impact of water-saving irrigation practices on grapevines. Chou et al. [6] reported that maintaining the soil water content at 60 ± 5% of field capacity significantly increased the soluble solids and vitamin C content in Pinot Noir grapes, decreased titratable acidity, and enhanced overall fruit quality. Lei Jun et al. [7] demonstrated that an irrigation volume of 3600 m^3^/ha improved fruit quality of ‘*Marselan*’ grapes, identifying it as the more appropriate irrigation level. Wang Weijun et al. [8] reported that the optimal irrigation volume for the growth period of Chunguang grapes was 2400 m^3^/ha. Consequently, enhancing the drought tolerance and resistance of grapevines is of paramount importance.

Melatonin (MT) is an indole derivative commonly found in organisms. As a plant growth regulator, it can not only regulate diverse physiological processes during plant growth and development, such as anti-aging [9], seed germination [10], and stomatal aperture [11], but also respond to the adverse effects of abiotic stress. In particular, it has strong reactive-oxygen-species-scavenging ability and antioxidant properties, which can enhance the antioxidant defense system of plants under stress [12,13]. In recent years, there has been an increasing number of studies on the role of melatonin in alleviating plant stress. Shi Zhongfei et al. [14] demonstrated that exogenous melatonin treatment mitigated low-temperature stress-induced damage by enhancing the activities of superoxide dismutase (SOD) and peroxidase (POD) and decreasing malondialdehyde (MDA) content in rape seedlings. Under drought stress conditions, spraying melatonin on citrus leaves has been shown to augment the antioxidant activity and bolster drought resistance in citrus, thereby increasing the quality of citrus in arid and semi-arid regions [15]. Jahan et al. [16] found that melatonin can enhance the thermotolerance of tomato plants by decreasing reactive oxygen species (ROS) production, increasing the activity of antioxidant enzymes, modulating the ascorbic acid–glutathione (AsA–GSH) cycle, and upregulating the expression of genes associated with antioxidant defense in tomato seedlings. Studies have revealed that the foliar application of melatonin under low-temperature stress can significantly elevate the levels of the endogenous hormones indole-3-acetic acid (IAA), gibberellin 3 (GA_3_), and melatonin (MT) in watermelon [17] and cucumber seedlings [18], as well as reduce the level of abscisic acid (ABA), thereby improving its cold tolerance. However, at present, there are no reports on the effects of MT on antioxidant properties, the dynamics of endogenous hormone levels, and fruit quality in grapevines under varying irrigation amounts, and its regulatory mechanism is still unclear.

Consequently, in this study, ‘Red Globe’ grape varieties were used as experimental materials, and melatonin was applied at specific concentrations to the leaves under varying irrigation amounts to investigate the physiological and molecular mechanisms by which melatonin (MT) enhances grape antioxidant capacity, hormonal balance, and fruit quality under these different irrigation conditions. This study aims to provide a theoretical foundation and reference for the use of exogenous melatonin to alleviate water deficit damage, as well as provide technical support for the sustainable development of the grape industry in arid regions.

## 2. Results

### 2.1. Effect of Melatonin on Membrane Permeability in Grape Leaves Under Varying Irrigation Amounts

With the reduction in irrigation, the relative conductivity of leaves increased gradually from 5 to 65 days post-anthesis, followed by a fluctuating pattern of initial decrease and subsequent increase from 95 to 155 days post-anthesis. Melatonin treatment led to a significant reduction in relative conductivity under different irrigation amounts. Specifically, the W1MT, W2MT, W3MT, and W4MT treatments resulted in average decreases of 5.71%, 5.86%, 12.92%, and 10.40%, respectively, compared to the control treatments W1CK, W2CK, W3CK, and W4CK (Figure 1A). The content of free proline followed a trend of initially increasing and then decreasing from 5 to 65 days post-anthesis and vice versa from 95 to 155 days post-anthesis. The MT treatment significantly enhanced the Pro content under different irrigation regimes. Notably, the W1MT treatment demonstrated the most pronounced effect (*p* < 0.05), with increases of 20.21%, 4.10%, 6.76%, 13.52%, 3.97%, and 6.08% over the W1CK treatment in each respective period (Figure 1B). As shown in Figure 1C,D, the contents of soluble protein and soluble sugar increased progressively from 5 to 65 days post-anthesis and then exhibited a trend of a slight decrease followed by an increase from 95 to 155 days post-anthesis. The MT treatment also increased the leaf contents of soluble protein and soluble sugar under different irrigation amounts. The W1MT and W2MT treatments had the most obvious effects on the Pro content at 155 days post-anthesis, resulting in the greatest enhancements, which were 12.14% and 12.20% higher than W1CK and W2CK, respectively. The W3MT and W4MT treatments led to the greatest enhancements in the Pro content at 35 days post-anthesis, with increases of 6.65% and 5.88% compared to W3CK and W4CK, respectively. Concurrently, the soluble sugar content with the W1MT, W2MT, W3MT, and W4MT treatments increased by 5.81%, 7.77%, 5.66%, and 4.61%, respectively, compared to the corresponding control treatments.

### 2.2. Effect of Melatonin on Active Oxygen Levels in Grape Leaves Under Varying Irrigation Amounts

As depicted in Figure 2A–C, the levels of O_2_^−^, H_2_O_2_, and MDA in grape leaves increased with reduced amounts of irrigation. However, the O_2_^−^ and H_2_O_2_ levels decreased slightly under the W2CK treatment between 95 and 155 days post-anthesis. The MT treatment inhibited the accumulation of ROS and MDA in grape leaves under various irrigation amounts. Specifically, the W4MT treatment had the most significant decreases in O_2_^−^ levels (*p* < 0.05), which were 10.27%, 6.57%, 9.33%, 15.40%, 9.55%, and 11.00% lower than with the W4CK treatment in different periods. The W1MT, W2MT, W3MT, and W4MT treatments also significantly lowered the H_2_O_2_ content by 5.76%, 5.83%, 4.71%, and 5.15%, respectively, compared to their corresponding control treatments. In terms of the MDA content, the W3MT treatment had the most pronounced effect (*p* < 0.05), reducing levels by 16.33%, 6.83%, 8.07%, 6.06%, 12.70%, and 13.83% compared to the W3CK treatment, respectively. The W4MT treatment was the second most effective, with levels 10.56%, 7.03%, 5.48%, 10.03%, 14.91%, and 12.07% lower than that of W4CK, respectively.

### 2.3. Effects of Melatonin on the Activity of Protective Enzymes in Grape Leaves Under Varying Irrigation Amounts

As illustrated in Figure 3, the enzymatic activities of SOD (Figure 3A), POD (Figure 3B), and CAT (Figure 3C) in grape leaves exhibited a coherent trend with the progression of grape development, characterized by an overall decline. Concurrently, a reduction in irrigation volume was correlated with a gradual increase in the activities of these enzymes from 5 to 65 days post-anthesis, in the sequence W1CK < W2CK < W3CK < W4CK. Subsequently, during the period from 95 to 155 days post-anthesis, the enzymatic activities showed a pattern of initial decrease followed by an increase. Specifically, the activities of SOD, POD, and CAT with the W2CK treatment were notably lower, decreasing by 54.95%, 63.83%, and 82.04%; 32.60%, 31.34%, and 43.19%; and 54.77%, 47.16%, and 65.45%, respectively, compared to the W4CK treatment. Under varying irrigation amounts, the application of 150 μmol·L^−1^ MT treatment promoted the activities of SOD, POD, and CAT by different degrees. Specifically, the W3MT treatment had the most significant effect on SOD activity (*p* < 0.05), with levels 10.61%, 21.29%, 4.35%, 7.56%, 25.79%, and 18.18% higher than with W3CK in each period. The W4MT treatment had the most significant effects on the POD and CAT activities (*p* < 0.05). Compared to the W4CK treatment in each period, the activities increased by 12.02%, 20.68%, 11.30%, 15.08%, 10.29%, 6.28%, and 24.77% for POD and by 22.06%, 11.89%, 14.56%, 16.94%, and 15.02% for CAT.

### 2.4. Effect of Melatonin on Ascorbate–Glutathione Cycle Enzyme Activity in Grape Leaves Under Varying Irrigation Amounts

It can be seen from Figure 4 that the activities of AAO (Figure 4A), APX (Figure 4B) DHAR (Figure 4C), MDHAR (Figure 4D), and GR (Figure 4E) in grapevine leaves initially increased and then subsequently decreased with the advancement of the grape growth period, reaching a peak at 65 days post-anthesis. With a reduction in the irrigation volume, the activities of AAO, APX, DHAR, MDHAR, and GR in the leaves showed gradual increases. However, the W2CK treatment exhibited a decrease in enzyme activity between 125 and 155 days post-anthesis, with no significant difference observed compared to the W1CK treatment. The activities of AAO, APX, DHAR, MDHAR, and GR were significantly enhanced by the MT treatment under varying irrigation amounts in each period. Specifically, the activity of AAO with the W1MT, W2MT, W3MT, and W4MT treatments increased by 3.81%, 4.13%, 4.43%, and 4.33%, respectively, over those of the corresponding W1CK, W2CK, W3CK, and W4CK treatments. Similarly, compared with the W1CK, W2CK, W3CK, and W4CK treatments, the activity of APX with W1MT, W2MT, W3MT, and W4MT increased by 15.79%, 10.65%, 12.08%, and 12.30%, respectively. The DHAR activity with the W1MT, W2MT, W3MT, and W4MT treatments increased by 10.87%, 14.31%, 10.42%, and 9.37%, respectively, relative to the W1CK, W2CK, W3CK, and W4CK treatments. The MDHAR activity with the W1MT, W2MT, W3MT, and W4MT treatments increased by 5.34%, 5.14%, 6.98%, and 6.19%, respectively, when compared with the W1CK, W2CK, W3CK, and W4CK treatments. Lastly, the GR activity with the W1MT, W2MT, W3MT, and W4MT treatments increased by 6.64%, 6.66%, 7.98%, and 5.96%, respectively, compared with the W1CK, W2CK, W3CK, and W4CK treatments.

### 2.5. Effect of Melatonin on Ascorbic Acid–Glutathione Levels in Grape Leaves Under Varying Irrigation Amounts

Figure 5 depicts the fluctuating levels of AsA, DHA, GSH, GSSG, GSH/GSSG ratio, and AsA/DHA ratio in grape leaves, which initially rose and then declined over the grape development. With the decrease in the amount of irrigation, the levels of AsA, DHA, GSH, GSSG, GSH/GSSG ratio, and AsA/DHA ratio in grape leaves increased to varying degrees. The application of MT under the varying irrigation regimes significantly enhanced the levels of AsA, DHA, GSH, the GSH/GSSG ratio, and the AsA/DHA ratio, while the concentration of GSSG was markedly suppressed.

### 2.6. Effects of Melatonin on Endogenous MT Content and Related Gene Expression in Grape Leaves Under Varying Irrigation Amounts

As shown in Figure 6, when the amount of irrigation was reduced at various developmental stages, the endogenous MT content in the grape leaves gradually increased. Thus, the MT content with the W4CK treatment was the highest, increasing by 25.99%, 13.42%, 16.80%, 20.83%, 21.15%, and 30.31% at each respective stage compared with the W1CK treatment. Furthermore, following the application of 150 μmol·L^−1^ MT under different irrigation amounts, the endogenous MT content increased to varying degrees. The W1MT, W2MT, W3MT, and W4MT treatments led to increases of, on average, 8.27%, 6.06%, 7.47%, and 6.10% in the MT content compared with their corresponding control treatments (W1CK, W2CK, W3CK, and W4CK). This indicates that the exogenous application of MT can facilitate the accumulation of endogenous MT in leaves.

*TDC1*, *T5H1*, *SNAT1,* and *ASMT1* are pivotal genes in the biosynthesis of melatonin. Figure 7A shows that with a decrease in the irrigation amount, the relative expression levels of *VvTDC1* significantly increased at 5, 35, and 95 days post-anthesis, and the increases were relatively gentle at 65, 125, and 155 days post-anthesis. The application of exogenous melatonin (MT) under varying irrigation conditions led to a significant upregulation of *VvTDC1* expression, with the W4MT treatment showing the most pronounced effect (*p* < 0.05), with levels 22.39%, 44.64%, 187.12%, 104.65%, 642.96%, and 220.88% higher than that of the W4CK treatment in each respective period. Figure 7B demonstrates that with the decrease in the irrigation amount, the relative expression of *VvT5H1* gradually increased from 5 to 65 days post-anthesis, slightly decreased between 95 and 125 days post-flowering, and then gradually increased, but there was no significant change at 155 days post-flowering. The relative expression of *VvT5H1* was upregulated to varying degrees following the MT treatment compared with the control (CK) treatment under the same amount of irrigation, and the difference was as follows: W2MT > W1MT > W3MT > W4MT. Figure 7C,D reveal that the relative expression of *VvSNAT1* and *VvASMT1* increased with a decrease in the irrigation amount, although no significant difference was found in the relative expression of *VvSNAT1* between the W1CK and W2CK treatments during the late growth stage (i.e., 95 to 155 days post-anthesis). The exogenous application of MT significantly upregulated the relative expression of both *VvSNAT1* and *VvASMT1* under different irrigation amounts, and the upregulation with the W1MT treatment was the most significant (*p* < 0.05), with levels 82.03%, 84.01%, 12.59%, 131.75%, 124.99%, and 328.08% and 68.24%, 158.54%, 467.28%, 124.01%, 48.37%, and 50.73% higher than that of the W1CK treatment, respectively.

### 2.7. Effects of Melatonin on Endogenous IAA Content and Related Gene Expression in Grape Leaves Under Varying Irrigation Amounts

Figure 8 illustrates that the IAA content in grape leaves gradually declined with the decrease in the amount of irrigation. Notably, the W4CK treatment resulted in the lowest IAA levels, decreasing by 24.44%, 24.37%, 25.46%, 22.83%, 32.18%, and 35.17% compared with the W1CK treatment in each period. Moreover, the exogenous application of MT significantly promoted the accumulation of IAA under varying irrigation amounts, and the increase was particularly effective (*p* < 0.05) with the W4MT treatment, with levels 6.29%, 3.44%, 6.61%, 12.38%, 21.31%, and 20.44% higher than that of the W4CK treatment in each stage. A comparison of the IAA content with the W1MT, W2MT, W3MT, and W4MT treatments relative to their control counterparts (W1CK, W2CK, W3CK, and W4CK) revealed, on average, increases of 7.03%, 9.52%, 6.22%, and 11.29%, respectively.

*YUCCA6*, *AUX1,* and *TIR* are pivotal genes involved in the biosynthesis and signal transduction of IAA. Figure 9 indicates that the relative expression levels of *VvYUCCA6*, *VvAUX1,* and *VvTIR* were downregulated with decreasing irrigation amounts. Conversely, the relative expression levels of *VvYUCCA6*, *VvAUX1,* and *VvTIR* were upregulated by the MT treatment under varying irrigation conditions. Among them, the W1MT treatment had the most significant upregulation effect on *VvYUCCA6* (*p* < 0.05), with increases of 30.30%, 316.90%, 154.41%, 233.39%, 192.08%, and 126.78% compared with the W1CK treatment at each respective time point. Furthermore, the W4MT treatment was particularly effective in upregulation of *VvAUX1* and *VvTIR*, which increased by 24.47%, 42.19%, 51.35%, 85.57%, 98.66%, and 103.26% and 32.24%, 54.29%, 239.02%, 29.06%, 25.94%, and 68.23%, respectively, compared with the W1CK treatment.

### 2.8. Effects of Melatonin on Endogenous ZT and TZR Contents and Related Gene Expression in Grape Leaves Under Varying Irrigation Amounts

ZT and TZR are naturally occurring plant cytokinins found in higher plants. As shown in Figure 10, the leaf contents of ZT and TZR decreased as the irrigation amount was reduced. The MT treatment promoted the accumulation of both ZT and TZR under varying irrigation amounts. Notably, the effect of the W4MT treatment was the most significant (*p* < 0.05), with ZT and TZR contents increasing by 4.89%, 11.45%, 8.54%, 13.35%, 13.10%, and 17.48% and 17.92%, 12.27%, 18.42%, 24.19%, 41.44%, and 41.32%, respectively, compared with the W4CK treatment at each corresponding stage. Furthermore, the contents of ZT and TZR with the W1MT, W2MT, W3MT, and W4MT treatments were 5.14%, 9.19%, 7.69%, and 10.73% and 9.22%, 12.44%, 15.33%, and 21.36% higher than those with the W1CK, W2CK, W3CK, and W4CK treatments, respectively.

*IPT2* and *CKX1* are key regulatory genes involved in cytokinin synthesis and degradation, respectively. Figure 11A,B show that the relative expression of *VvIPT2* and *VvCKX1* decreased with the reduction in the irrigation amount. The application of MT significantly upregulated the expression of both *VvIPT2* and *VvCKX1* under different irrigation amounts, and the W4MT treatment was the most significant (*p* < 0.05), which was 151.11%, 59.99%, 33.07%, 83.07%, 114.38%, and 85.61% and 11.44%, 106.18%, 46.46%, 24.19%, 36.93%, and 33.15% higher than that of W4CK in each period, respectively.

### 2.9. Effects of Melatonin on Endogenous GA_3_ Content and Related Gene Expression in Grape Leaves Under Varying Irrigation Amounts

Figure 12 shows that when the amount of irrigation was reduced, the content of GA_3_ in the grape leaves also tended to decrease. However, the content of GA_3_ with the W2CK treatment notably increases between 95 and 155 days post-anthesis. The application of 150 μmol·L^−1^ MT treatment significantly enhanced the content of GA_3_ to different degrees under the various irrigation amounts. The W1MT, W2MT, W3MT, and W4MT treatments resulted in respective average increases of 9.68%, 9.13%, 7.90%, and 10.01% compared with the W1CK, W2CK, W3CK, and W4CK treatments.

Figure 13 illustrates that the relative expression of *VvG20ox1*, a gene associated with gibberellin biosynthesis, decreased with the reduction in irrigation, whereas the expression of *VvDELLA*, a repressor of gibberellin signaling, increased. The application of MT significantly modulated these expressions under different irrigation amounts; on average, it upregulated *VvG20ox1* by 59.31% in W1MT and 69.52% in W4MT compared with their respective controls (W1CK and W4CK). Conversely, the MT treatment, on average, downregulated *VvDELLA* expression by 40.80% in W1MT and 33.23% in W4MT relative to W1CK and W4CK.

### 2.10. Effects of Melatonin on Endogenous ABA Content and Related Gene Expression in Grape Leaves Under Varying Irrigation Amounts

ABA plays a crucial physiological role in plant response to biotic and abiotic stresses. Figure 14 shows that the ABA in the leaves gradually accumulated when the irrigation amount decreased; thus, the accumulation of ABA was the highest under the W4CK treatment. The MT treatment significantly reduced the content of ABA in leaves under different irrigation amounts, with levels for the W1MT, W2MT, W3MT, and W4MT treatments decreasing by 11.77%, 11.63%, 11.54%, and 11.45% on average compared with the W1CK, W2CK, W3CK, and W4CK treatments.

*NCED1*, *PP2C*, *SNRK2*, and *ABAH3* are pivotal genes implicated in biosynthesis, signal transduction, and degradation of ABA. Figure 15 illustrates that as the irrigation amount decreased, the expression of *VvNCED1* increased, whereas the relative expression of *VvPP2C*, *VvSNRK2*, and *VvABAH3* decreased. For the MT treatment under the different irrigation amounts, the relative expression of *VvNCED1* was notably downregulated, with the W1MT treatment exhibiting the most significant reduction, being 46.16% lower than that of the corresponding W1CK treatment for each period. *PP2C*, as a negative regulator in ABA signal transduction, was upregulated in its relative expression upon the MT treatment, with the W1MT treatment being particularly significant (*p* < 0.05). Specifically, the upregulation was 72.75%, 29.42%, 59.82%, 456.12%, 128.91%, and 120.36% at different stages compared with the W1CK treatment. Conversely, *VvSNRK2*, as the corresponding positive regulator, and the MT treatment downregulated its relative expression. Additionally, the MT treatment induced an upregulated expression of *VvABAH3*, with the W1MT and W4MT treatments showing average increases of 87.44% and 45.57%, respectively, when compared with the W1CK and W4CK treatments.

### 2.11. Effects of Melatonin on Endogenous SA Content and Related Gene Expression in Grape Leaves Under Varying Irrigation Amounts

Figure 16 demonstrates that as the grape growth period advanced, the SA content in the grape leaves showed a trend of ‘initially increasing and subsequently decreasing’, reaching the peak at 95 days post-anthesis. MT treatment was found to reduce the SA content in leaves under different irrigation amounts, indicating that MT was beneficial to the metabolic decomposition of SA. Among them, the W3MT treatment resulted in the most significant reductions (*p* < 0.05), which were 3.17%, 5.86%, 19.71%, 5.18%, 13.11%, and 8.35% lower than those with the W3CK treatment in each period. The W4MT also exhibited a notable decrease, being 8.00% lower than with the W4CK treatment in each period.

### 2.12. Effect of Melatonin on Soluble Solids and Titratable Acid Content of Grape Berries Under Varying Irrigation Amounts

As presented in Table 1, with the decrease in the irrigation amount, the soluble solids, solid–acid ratio, and firmness of the grape berries first increased and then decreased. Conversely, the titratable acid content showed a trend of an initial decrease and subsequent increase. The MT treatment enhanced the soluble solids content, solid–acid ratio, and hardness of fruit under varying irrigation amounts while also curbing the accumulation of titratable acids. Among them, the W1MT treatment resulted in a significant elevation in the soluble solids content by 6.77% compared with the W1CK treatment. The W4MT treatment markedly increased the solid–acid ratio by 10.42% over the W4CK treatment; the W3MT treatment exerted the most pronounced effect on the fruit hardness and titratable acid, with respective increases and decreases of 22.22% and 7.41% compared with the W3CK treatment.

### 2.13. Effects of Melatonin on the Contents of Total Soluble Sugar and Sugar Components in Grape Berries Under Varying Irrigation Amounts

Table 2 indicates that with the decrease in the irrigation amount, the contents of total soluble sugar, glucose, fructose, and sucrose in the grape berries increased gradually. The application of MT treatment further elevated these sugar levels under different irrigation amounts. Among them, the increases in the contents with the W4MT treatment were the most significant, being 26.96%, 36.22%, 43.73%, and 17.31% higher than those with the W1MT treatment.

### 2.14. Effects of Melatonin on the Contents of Phenolic Compounds and Vitamin C in Grape Berries Under Varying Irrigation Amounts

The results in Table 3 show that the decrease in the amount of irrigation caused the contents of total phenols, total flavonoids, and vitamin C in the grape berries to initially increase and then decrease, as well as the tannin content to decrease gradually. The contents of total phenols and total flavonoids reached their peaks under the W3CK treatment, which were 6.03 and 2.42 mg·g^−1^, respectively. The accumulation of vitamin C was the highest under the W2CK treatment, which was 11.63 mg·100 g^−1^. Under different irrigation amounts, the MT treatment promoted the accumulation of total phenols, total flavonoids, and vitamin C in the grape berries, as well as accelerated the decomposition of tannins. Among them, the W1MT treatment had the most significant effect on the total phenol content (*p* < 0.05), which was 8.92% higher than that with the W1CK treatment. The W3MT treatment had the most obvious inhibitory effect on the fruit tannin content, which was 9.78% lower than that with the W3CK treatment. The W2MT treatment was particularly effective in elevating the total flavonoid content, showing an 18.89% increase over the W2CK treatment. However, the MT treatment did not significantly influence the increase in the vitamin C content.

### 2.15. Effects of Melatonin on Single Fruits’ Weight and Vertical and Transverse Diameters in Grape Berries Under Varying Irrigation Amounts

It can be seen from Table 4 that with the decrease in the irrigation amount, the weight and both vertical and transverse diameters of single grape berries initially increased and then decreased, indicating that the appropriate irrigation amount (W2CK) enhanced the fruit quality and berry size. Compared with the CK group, the MT treatment significantly augmented the weight and vertical and transverse diameters of single fruits. Specifically, the single fruit weights with the W1MT, W2MT, W3MT, and W4MT treatments were 13.89%, 13.27%, 7.98%, and 7.24% higher than those with the W1CK, W2CK, W3CK, and W4CK treatments, respectively. The vertical diameter increased by 4.80%, 4.06%, 4.62%, and 3.64%, while the transverse diameter increased by 5.04%, 4.63%, 4.72%, and 4.89%, respectively. The fruit shape index for each treatment exceeded 1 and increased as the irrigation amount decreased, with the MT treatment showing indices closer to 1 compared with the CK group.

### 2.16. Principal Component Analysis

As shown in Table 5, through a comprehensive analysis of the principal components for the physiological indicators of grape leaves, four principal components with eigenvalues greater than 1 were ultimately extracted (FAC1, FAC2, FAC3, and FAC4). The comprehensive score is the sum of the product of each principal component score and the corresponding contribution rate, namely, *F* = 72.83 FAC1 + 16.27 FAC2 + 6.78 FAC3 + 2.67 FAC4. The comprehensive scores of the different treatments were ranked, from high to low, as follows: W4MT > W3MT > W4CK > W3CK > W2MT > W2CK > W1MT > W1CK. This indicates that under the irrigation amount of 180mm (W4), the foliar application of 150 μmol·L^−1^ MT had the best effect on the antioxidant properties and endogenous hormones of the leaves.

## 3. Discussions

### 3.1. Impact of Melatonin on the Resistance Physiology in Grape Leaves Under Varying Irrigation Amounts

Leaf relative electrical conductivity (REC) is an important parameter for evaluating plant cell membrane permeability [19]. Malondialdehyde (MDA), a product of lipid peroxidation in plant cell membranes, serves as a marker for assessing cell membrane damage [20]. Free proline (Pro), soluble protein (SP), and soluble sugar (SS) are crucial osmotic regulators in plants and can be used to measure drought resistance [21]. Research has demonstrated that melatonin significantly reduces electrolyte leakage in kiwifruit seedlings under drought stress, thereby mitigating cell membrane damage [22]. Ye et al. reported that melatonin treatment decreases the malondialdehyde content in maize leaves under drought stress [23]. Qi Xiaoyuan also observed that melatonin can significantly increase the levels of osmotic adjustment substances in chrysanthemum under high-temperature stress [24]. Our research reveals that as the irrigation amount decreased, the leaf contents of REC, Pro, SP, SS, and MDA generally increased. However, the MT treatment significantly reduced the levels of REC and MDA under various irrigation amounts and significantly increased the contents of Pro, SP, and SS; this indicates that exogenous MT is beneficial for the accumulation of osmotic adjustment substances under drought stress, which reduces its cellular osmotic potential, maintaining cell turgor and water retention capacity and, thus, alleviating the drought-induced damage to grapevine leaf cell membranes to a certain extent [11].

Reactive oxygen species (ROS) are byproducts of normal metabolic processes in plants and play a crucial role in maintaining the intracellular environmental balance and participating in signal transduction pathways [25]. However, excessive ROS levels can cause serious damage to cellular structures [26]. With the reduction in irrigation, the leaf contents of O_2_^−^ and H_2_O_2_ increased significantly. Compared with the control treatment (CK) under the same amount of irrigation, the levels of O_2_^−^ and H_2_O_2_ were lower with the MT treatment, suggesting that exogenous melatonin can scavenge excessive ROS in plants, thereby reducing oxidative damage and enhancing their antioxidant capacity [27]. In this process, the synergistic action of superoxide dismutase (SOD), peroxidase (POD), and catalase (CAT) repairs the damage caused by reactive oxygen species and prevents membrane lipid peroxidation induced by stress [28]. Specifically, SOD catalyzes the disproportionation of O_2_^−^ into H_2_O_2_ and O_2_^−^, while CAT and POD further decompose H_2_O_2_ into H_2_O, protecting cells from its toxicity [29]. Related studies have demonstrated that melatonin can enhance the activity of antioxidant protective enzymes (SOD, POD, and CAT) in apple [30] and coffee [31] leaves under drought stress. In our experiment, we observed that the activities of SOD, POD, and CAT in leaves increased progressively with decreasing irrigation amounts, and the MT treatment significantly elevated the activities of these enzymes under various irrigation amounts. Therefore, exogenous MT may enhance the ROS scavenging ability of cells by activating antioxidant protective enzyme activities, thereby improving the tolerance of grape leaves to water stress.

In addition, the ascorbate–glutathione (AsA−GSH) cycle is a vital component of the non-enzymatic antioxidant system in plants and plays key roles in the reduction and regeneration of AsA in vivo [32]. It primarily consists of ascorbic acid (AsA), glutathione (GSH), dehydroascorbic acid (DHA), oxidized glutathione (GSSG), ascorbate oxidase (AAO), monodehydroascorbate reductase (MDHAR), dehydroascorbate reductase (DHAR), ascorbate peroxidase (APX), and glutathione reductase (GR) [33]. Within this cycle, AsA, GSH, DHA, and GSSG are significant antioxidants, while AAO, APX, MDHAR, DHAR, and GR are the principal antioxidant enzymes that collaborate to eliminate excess ROS accumulated under stress, mitigate cellular oxidative damage, and preserve normal metabolic processes [34]. Research has established that the AsA/DHA and GSH/GSSG ratios are direct indicators of a plant’s antioxidant capacity [35]. Under drought stress, melatonin treatment enhances the GR activity, which stimulates the synthesis of AsA and GSH, thereby increasing the AsA/DHA and GSH/GSSG ratios [30]. Ding et al. observed significant increases in the levels of AsA, DHA, GSH, and GSSG in potato leaves under drought stress [36]. The findings of this experiment revealed that as the irrigation decreased, the activities of AAO, APX, DHAR, MDHAR, and GR in leaves progressively increased, and the levels of AsA, GSH, DHA, and GSSG also rose significantly. The MT treatment significantly elevated the activities of these enzymes and increased the AsA/DHA and GSH/GSSG ratios in leaves under various irrigation amounts, indicating that under drought stress, melatonin can promote the accumulation of AsA and GSH in leaves by enhancing the activities of enzymes associated with the AsA-GSH cycle, thereby inhibiting ROS production and bolstering the plant’s antioxidant defense [37].

### 3.2. Impact of Melatonin on Endogenous Hormonal Dynamics in Grape Leaves Under Varying Irrigation Amounts

Plant endogenous hormones are natural organic substances that regulate physiological functions and growth rhythms in plants [38], and they are also key endogenous factors for plants to resist stress [39]. Under drought stress, the levels of endogenous hormones in plants change significantly, and the stress response depends not only on the regulation of a single hormone but also on the interactions among various hormones [40]. IAA is the most common plant growth regulator, playing vital roles in growth and development, as well as in resistance to environmental stress [41]. ABA is a key stress signal hormone that accumulates under plant stress. It can activate key enzymes in the ABA biosynthesis pathway, forming a positive feedback mechanism that enhances the ABA signal transduction pathway, thereby enabling plants to better adapt to stressful environments [42]. CTK and GA_3_, as corresponding negative regulatory signals, are preferentially perceived by cells when plants are under stress, promoting the synthesis of a large number of positive regulatory signals, such as ABA, which in turn weakens cell growth and metabolic activities [43]. SA is a phenolic hormone widely present in plants and plays significant roles in responses to biotic and abiotic stresses [44]. The results of this experiment showed that as the irrigation amount decreased, the leaf contents of endogenous IAA, GA_3_, ZT, and TZR gradually decreased, while the contents of MT, ABA, and SA gradually accumulated, consistent with the findings of Zhang et al. on the dynamic changes in endogenous hormones in sweet potatoes under drought stress [45]. Melatonin is also an important plant hormone and has complex interactions with other plant endogenous hormones (IAA, ABA, CTK, GA, and SA). Studies have found that exogenous melatonin can regulate changes in the endogenous hormone contents in plants to enhance their stress resistance [46]. Li Dong et al. found that exogenous melatonin positively affected the accumulation of IAA, GA_3_, and ZR in tobacco seedlings under drought stress and reduced the ABA content, thus enhancing the drought resistance of tobacco seedlings [39]. In this study, foliar application of MT increased the contents of IAA, ZT, TZR, and GA_3_ and inhibited the accumulation of ABA and SA under different irrigation levels. This result is consistent with the findings of Yang Yan et al. on crabapple, indicating that MT positively regulates plant endogenous hormones [47].

*TDC1*, *T5H1*, *SNAT1*, *ASMT1*, and *COMT1* are the principal regulatory genes in the MT biosynthesis pathway [48]. The *q*PCR results from this experiment revealed that the MT treatment significantly upregulated the expression of *VvTDC1*, *VvSNAT1*, *VvT5H1*, and *VvASMT1* under various irrigation conditions, consistent with the findings of Zheng et al. This suggests that exogenous MT can stimulate the expression of key genes involved in endogenous MT biosynthesis [49]. *YUCCA6* is a key gene in the IAA biosynthetic pathway, and *AUX1* encodes a transmembrane protein responsible for transporting auxin from the plasma membrane into the cell, with the upregulation of their expression promoting IAA synthesis in leaves [50]. Concurrently, *TIR* encodes an auxin receptor that interacts with IAA and Aux/IAA proteins to activate auxin-synthesis-related genes [51]. Research has indicated that melatonin can regulate the upregulation of *VvAUX1*-related differential gene expression in the auxin sub-pathway, which is beneficial for plant growth and enhances its drought resistance [52]. The findings of this experiment demonstrated that the MT treatment upregulated the expression of *VvYUCCA6*, *VvAUX1*, and *VvTIR* under different irrigation amounts, with the most pronounced effect observed under the W4MT treatment. This could be attributed to the ability of melatonin to activate the expression of genes related to IAA biosynthesis and signal transduction pathways, thereby promoting IAA accumulation. *IPT2* and *CKX1* are key genes involved in the CTK synthesis and degradation. With the reduction in irrigation, the expression of *VvIPT2* and *VvCKX1* was downregulated, suggesting that water stress inhibits the expression of genes related to ZT and TZR synthesis and degradation. After the MT treatment, the expression of *VvIPT2* and *VvCKX1* was significantly upregulated compared with the control (CK) treatment under the same irrigation conditions, indicating that MT can regulate the synthesis and degradation of ZT- and TZR-related genes, thereby increasing the leaf contents of ZT and TZR. GA_3_ regulate stomatal opening and closing under plant drought stress, reducing leaf transpiration. The antagonism between GA_3_ and ABA is mediated by the *DELLA* protein, which is the key mechanism for plants to avoid early abiotic stress [53]. Studies have shown that stress stimulates the activity of GA 2ox, leading to 2β-hydroxylation and activation of *DELLA*, resulting in GA inactivation [54]. In this study, the expression of *VvDELLA* was significantly downregulated after the MT treatment under various irrigation amounts, and the leaf content of GA_3_ significantly increased. It is hypothesized that the GA signal depends on the degradation of the *DELLA* protein mediated by ubiquitination, and MT may enhance the GA_3_ signal in grapes by reducing the expression of *VvDELLA*, thus facilitating the response to abiotic stress. *NCED1,* encoding 9-cis-epoxycarotenoid dioxygenase, is a key rate-limiting enzyme in ABA biosynthesis. *PP2C* and *SnRK2* are negative and positive regulators, respectively, in the ABA signal transduction pathway and are involved in plant drought stress response. *ABAH3* encodes ABA 8’-hydroxylase, a key enzyme that catalyzes ABA catabolism [27,55]. Chao L. found that melatonin treatment downregulated the expression of the ABA biosynthesis gene *MdNCED3* and upregulated the expression of the catabolic genes *MdCYP707A1* and *MdCYP707A2* in apple leaves under drought stress [27]. Haitao S. found that melatonin can induce the expression of *PP2C* and downregulate the expression of *SnRK2* under abiotic stress through transcriptome analysis [56]. The results of this experiment showed that the expression of *VvNCED1* and *VvSNRK2* increased with a decrease in the irrigation. After the MT treatment under the different irrigation amounts, the expression of *VvNCED1* and *VvSNRK2* was significantly downregulated, while the expression of *VvABAH3* and *VvPP2C* was upregulated. Therefore, it is suggested that MT may induce the expression of *VvABAH3* and *VvPP2C* to accelerate ABA degradation by downregulating the expression of *VvNCED1* and *VvSnRK2*, thereby enhancing resistance to drought stress.

### 3.3. Impact of Melatonin on the Quality of Grape Berries Under Varying Irrigation Amounts

Soluble sugar, soluble solids, titratable acid, and phenolic compounds are crucial indicators for assessing the intrinsic quality of fruits. Grape berries are classified as the hexose accumulation type; hexoses are primarily metabolized before ripening and accumulate mainly after ripening [57]. Consequently, the reducing sugars in the fruit are predominantly derived from glucose and fructose, with lower levels of sucrose [58]. Lin et al. demonstrated that moderate drought stress during the maturity stage in grapes effectively promoted the accumulation of glucose, fructose, and sucrose in the fruits [59]. Yang et al. discovered that the contents of glucose and fructose in Cabernet Sauvignon grape berries were higher under reduced irrigation (60% of the normal irrigation amount) compared with normal management conditions. Real-time fluorescence quantitative analysis revealed that the transcription levels of monosaccharide transporters and fructosidase, which are related to sugar accumulation, were upregulated to varying degrees under drought stress [60]. This study found that a decrease in the irrigation amount significantly increased the contents of glucose, fructose, and sucrose in the fruits (*p* < 0.05). These increases are attributed to water stress promoting the translocation of assimilates produced by leaf photosynthesis to the fruits, thereby significantly enhancing the accumulation of berry sugars. Titratable acidity is also one of the critical factors influencing grape quality. This study found that a reduction in the irrigation amount inhibited the accumulation of titratable acidity. Compared to the control under full irrigation (W1CK), the differences in the titratable acidity under reduced irrigation conditions (W2CK, W3CK, and W4CK) were statistically significant (*p* < 0.05), which is consistent with the findings of He. A. [61]. In addition, the types and contents of vitamin C and phenolic compounds are also particularly important for fruit quality. Phenolics are secondary metabolites in grape berries, which include tannins, phenolic acids, flavonoids, and anthocyanins, predominantly found in the skin and seeds of grapes. Research has indicated that moderate water stress can increase the vitamin C content in fresh grape berries by 15.3% to 42.2% and also enhance the tannin content in grape skins [62]. The findings of this study revealed that reducing the irrigation amount led to varying degrees of increases in the vitamin C content of the fruit and the levels of total phenols, total flavonoids, and tannins in the peel. Prior studies have demonstrated that exogenous melatonin treatment can effectively enhance the quality of grape [63], cherry [64], and papaya [65] fruits under drought stress. This study’s results showed that applying MT under various irrigation amounts increased the soluble solids, total soluble sugar, sugar components, solid–acid ratio, firmness, and vitamin C contents of the fruit, as well as the total phenol and total flavonoid contents in the peel, while decreasing the titratable acid and tannin contents. The underlying reason may be that melatonin can enhance leaf photosynthetic capacity, thereby promoting the transport and transformation of carbohydrates and, ultimately, improving fruit quality [66,67]. Moreover, it is worth mentioning that foliar application of melatonin significantly increased in the longitudinal and latitudinal diameters of the fruit, as well as the weight of the fruit berries, etc.

## 4. Materials and Methods

### 4.1. Overview of the Test Site and Test Materials

The experiment was conducted from May to November 2023 in a solar greenhouse at the grape demonstration base in Huangyang Town, Wuwei City (E 101°59′ to 103°23′, N 37°23′ to 38°12′). The site is situated at an altitude of approximately 2000 m and belongs to the temperate continental arid climate; this region experiences four distinct seasons, with sufficient sunshine and a large diurnal temperature variation. The precipitation during the growing season is approximately 97.4 mm, while the annual evaporation is approximately 1000 to 2000 mm. The average annual temperature is around 7.7 °C, with a total of approximately 2873 h of sunshine per year.

The irrigation method employed was drip irrigation under film. A drip irrigation belt was installed on both sides of each row of grapevines. The diameter of the drip holes was 16 mm, with 20 drip holes per strip. The flow rate was 1.6 L·h^−1^. Other than this, standard field management practices were followed.

The test variety was a 7-year-old ‘Red Earth’ table grape cultivar, with a plant spacing of 1 m × 2 m, cultivated as a single-wall hedge in a north–south orientation. The melatonin (MT) used in the test was purchased from Shanghai (China) yuanye Bio-Technology Co., Ltd., with a purity of 99%.

### 4.2. Experimental Design

A total of four irrigation treatments were established for the experiment, as follows: 360 mm (W1), 300 mm (W2), 240 mm (W3), and 180 mm (W4). The leaves were sprayed with 150 μmol·L^−1^ MT under varying irrigation amounts at each growth stage, with water sprayed as the control (CK). There were eight treatment combinations, as follows: W1CK, W1MT, W2CK, W2MT, W3CK, W3MT, W4CK, and W4MT. The spraying was conducted at 19:00 on days 0, 30, 60, and 90 post-anthesis. The spraying standard was that the entire plant’s leaves were completely wet and dripping. Four adjacent rows were randomly selected as one plot, with protective rows in between. Each irrigation treatment was replicated three times, resulting in a total of 12 plots.

Leaves were sampled at 5, 35, 65, 95, 125, and 155 days post-anthesis. Samples were collected from leaves located at 5 to 7 nodes near the base of the main shoots. Fruit sampling was conducted at 155 days post-anthesis (the mature period), with one fruit cluster randomly harvested from each plant per treatment. The samples were immediately frozen in liquid nitrogen, then stored in dry ice for transportation to the laboratory, and subsequently placed into a −80 °C ultra-low-temperature freezer.

### 4.3. Determination Items and Methods

#### 4.3.1. Determination of Relative Conductivity and Osmotic Adjustment Substance Contents in Leaves

Th relative conductivity (REC) was determined by the method of Zhou [68]; free proline (Pro) content was determined using the method of Li Yongzhen [69]; and the contents of soluble sugar (SS) and soluble protein (SP) were determined according to Wang’s method [70].

#### 4.3.2. Determination of the Active Oxygen Level and Malondialdehyde Content in Leaves

The content of H_2_O_2_ and the rate of O_2_^−^ production were determined using Jeannette’s method [71]. The malondialdehyde (MDA) content was determined using the method of Wu [72].

#### 4.3.3. Determination of the Antioxidant-Related Enzyme Activity in Leaves

Superoxide dismutase (SOD) activity was determined using the nitro blue tetrazolium (NBT) assay [73], with the enzyme activity expressed as one unit (U) defined as the amount required to achieve 50% inhibition of NBT photochemical reduction. Peroxidase (POD) activity was determined using the guaiacol method at 470 nm [74], with one unit (U) defined as the amount of enzyme catalyzing the decomposition of 1 μmol of substrate (guaiacol) per minute. Catalase (CAT) activity was calculated by monitoring the change in H_2_O_2_ at 240 nm [75], with one unit (U) defined as the amount of enzyme catalyzing the decomposition of 1 μmol of H_2_O_2_ per minute.

#### 4.3.4. Ascorbic Acid–Glutathione Cycle Antioxidant-Related Enzyme Activity Determination in Leaves

The activities of ascorbate oxidase (AAO), ascorbate peroxidase (APX), dehydroascorbate reductase (DHAR), monodehydroascorbate reductase (MDHAR), and glutathione reductase (GR) were measured following the method of Ramzi Murshed [76]. The enzyme activity was defined as the rate of absorbance change, with an increment of 0.01 absorbance units per minute constituting one unit of enzyme activity (U).

#### 4.3.5. Determination of the Ascorbic Acid–Glutathione Cycle Antioxidant Content in Leaves

The ascorbic acid (AsA) and dehydroascorbic acid (DHA) contents were quantified using the analytical procedure described by Kelly M. Gillespie [77]. The reduced glutathione (GSH) and oxidized glutathione (GSSG) levels were assessed employing the 5,5′-dithiobis (2-nitrobenzoic acid) (DTNB) assay, as detailed in the literature [78].

#### 4.3.6. Determination of the Endogenous Hormone Contents in Leaves

The endogenous hormone contents of melatonin (MT), indoleacetic acid (IAA), zeatin (ZT), trans-zeatin (TZR), gibberellin (GA_3_), abscisic acid (ABA), and salicylic acid (SA) were determined by the LC–MS/MS method [79]. Quantification was performed using an Agilent 1100 high-performance liquid chromatography (HPLC) system (Agilent Technologies, Santa Clara, CA, USA) equipped with a ZORBAX SB-C18 column (4.6 × 250 mm, 5 µm). The chromatographic conditions were as follows: column temperature maintained at 30 °C, a mobile phase composed of methanol and 0.1% phosphoric acid (volume ratio of 19), a flow rate of 1.0 mL·min^−1^, detection at a wavelength of 254 nm, and an injection volume of 10 µL.

#### 4.3.7. Real-Time Fluorescence Quantitative *q*RT–PCR

Total RNA was extracted from grapevine leaves using the RNA prep Pure Polysaccharide Polyphenol Plant Total RNA Kit. Following the RNA extraction, complementary DNA (cDNA) was synthesized using the Prime Script RT Reagent Kit with gDNA Eraser. The cDNA was then diluted to a concentration of 100 ng·μL^−1^ to serve as the template for the quantitative polymerase chain reaction (*q*PCR). The genes related to the melatonin synthesis and endogenous hormones in grape leaves were screened through the NCBI database, and the primers were designed using and synthesized by Sangon Biotech (Shanghai, China) Co., Ltd. The primer sequences are shown in Table 6.

The *q*PCR reaction system was as follows: 2 × SuperReal PreMix Plus 10 μL, upstream and downstream primers of 0.4 μL each, cDNA template of 3 μL, dd H_2_O of 6.2 μL, 3 replicates for each sample, and 3 technical replicates. The *q*PCR reaction was programmed as follows: pre-denaturation at 95 °C for 30 s, denaturation at 95 °C for 15 s, annealing at 60 °C for 30 s, and 40 cycles. The PCR reaction program was set as follows: pre-denaturation at 95 °C for 30 s, denaturation at 95 °C for 15 s, annealing at 60 °C for 30 s, and a total of 40 cycles. The melting curve was set to the default setting of the instrument, which lasted for 10 s at 95 °C, 60 s at 65 °C, and 1 s at 97 °C. The relative expression levels of the genes were calculated using the formula of 2^−∆∆CT^.

#### 4.3.8. Determination of the Fruit Quality and Yield-Related Indicators

The soluble solids (TSS) contents were measured using a PAL-1 digital refractometer (ATAGO CO., LTD., Tokyo, Japan). The content of titratable acid was determined by sodium hydroxide titration [80]. The solid–acid ratio is expressed as the ratio of soluble solids to titratable acid; and the content of soluble sugar (berry) was determined by anthrone colorimetry. The contents of glucose, fructose, and sucrose were determined by high-performance liquid chromatography (Waters Acquity Arc, Milford, MA, USA). The determination method was based on that of He Yajuan [81]. The HPLC conditions were optimized using an XBridge BEH Amide column (4.6 × 150 mm, 2.5 μm), with a column temperature set at 40 °C; a mobile phase composed of 75% acetonitrile, 0.2% ethylamine, and 24.8% ultrapure water; a flow rate of 0.8 mL·min^−1^; an injection volume of 10 μL; and a detection wavelength of 254 nm. The total phenolic content was determined by the Folin–Ciocalteu reagent method [82]. The content of tannin was determined by the Folin–Denis reagent method [83]. The total flavonoid content was quantified according to the aluminum chloride colorimetric method [84]. Fruit berry weights were determined using an electronic balance with a precision of 0.001 g. The vertical and transverse diameters were measured using a vernier caliper.

### 4.4. Statistical Analysis of Data

SPSS 23.0 was used for data processing, and Duncan’s multiple comparisons was used for analysis of the variance. Excel 2021 and Origin 2022 were used for drawing the charts.

## 5. Conclusions

The findings of our study indicate that the application of MT at different developmental stages of grapes under varying irrigation amounts significantly increased the content of osmotic adjustment substances in leaves, maintaining the osmotic balance and integrity of plant cell membranes. Concurrently, by reducing the levels of reactive oxygen species, enhancing the activity of antioxidant enzymes (SOD, POD, and CAT), as well as regulating the ASA-GSH cycle metabolism, we effectively mitigated the oxidative stress, strengthened the antioxidant defense mechanisms of grapes, and further alleviated the damage caused by the water stress to the grapes. In terms of endogenous hormone regulation, the application of MT under the different irrigation levels effectively increased the contents of MT, IAA, ZT, TZR, and GA_3_ in the grape leaves while promoting the degradation of ABA and SA, thus facilitating a balance among the plant hormones. It is particularly worth mentioning that the MT treatment also increased the contents of soluble solids, soluble sugars, sugar components (glucose, fructose, and sucrose), and total phenols in the grape berries, while reducing the contents of titratable acids and tannins, thereby improving the fruit quality. Although this study provides strong evidence for the role of melatonin in improving the physiological characteristics and fruit quality of grapes under different irrigation amounts, there are some limitations. For instance, while we have investigated the positive effects of MT on fruit quality, a detailed analysis of its specific impact on flavor quality has not been conducted. Future research will further explore the effects of the application of MT under different environmental conditions and analyze its impact on the flavor quality of grape berries, with the aim of providing a theoretical foundation for the application of MT in grape cultivation.

## Figures and Tables

**Figure 1 ijms-25-13081-f001:**
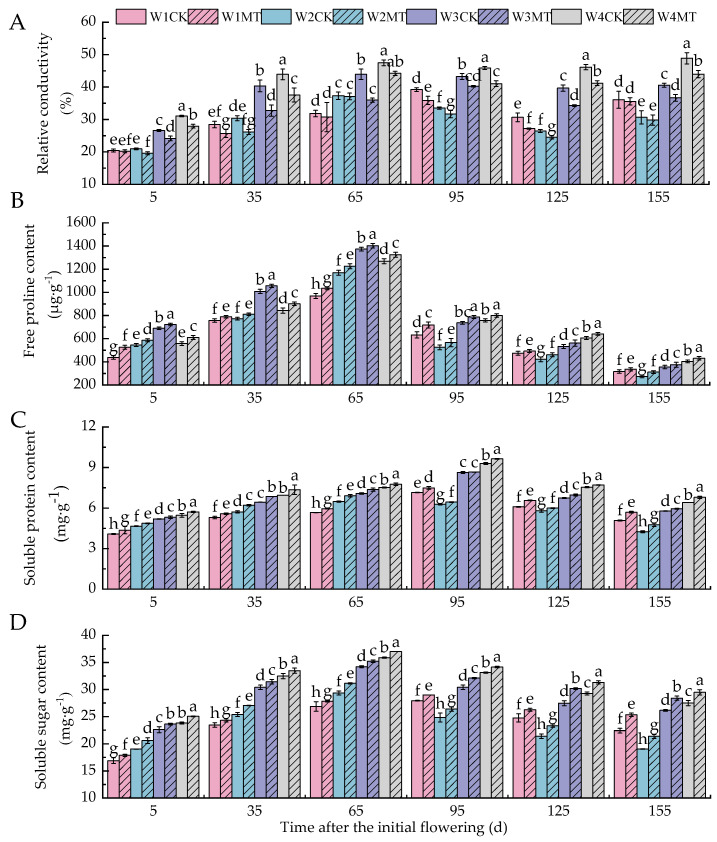
Effect of melatonin on the membrane permeability of grape leaves under varying irrigation amounts (values were determined based on the fresh weight): (**A**) relative conductivity (REC); (**B**) free proline (Pro); (**C**) soluble protein (SP); (**D**) soluble sugar (SS). W1CK (irrigated with 360 mm of water and water sprayed (instead of melatonin) on the leaves), W1MT (irrigated with 360 mm of water and 150 μmol·L^−1^ melatonin sprayed on the leaves), W2CK (irrigated with 300 mm of water and water sprayed (instead of melatonin) on the leaves), W2MT (irrigated with 300 mm of water and 150 μmol·L^−1^ melatonin sprayed on the leaves), W3CK (irrigated with 240 mm of water and water sprayed (instead of melatonin) on the leaves), W3MT (irrigated with 240 mm of water and 150 μmol·L^−1^ melatonin sprayed on the leaves), W4CK (irrigated with 180 mm of water and water sprayed (instead of melatonin) on the leaves), and W4MT (irrigated with 180 mm of water and 150 μmol·L^−1^ melatonin sprayed on the leaves). The results are the mean ± SE of three independent replications. Different letters represent significant differences among treatments (*p* < 0.05).

**Figure 2 ijms-25-13081-f002:**
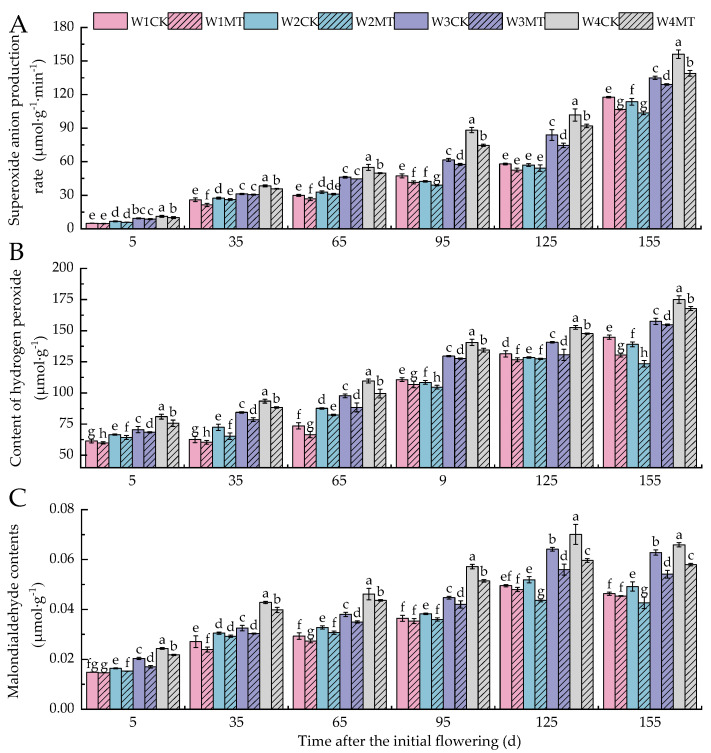
Effects of melatonin on the accumulation of reactive oxygen species and malondialdehyde in grape leaves under varying irrigation amounts (values were determined based on the fresh weight): (**A**) O_2_^−^; (**B**) H_2_O_2_; (**C**) malondialdehyde (MDA). W1CK (irrigated with 360 mm of water and water (instead of melatonin) sprayed on the leaves), W1MT (irrigated with 360 mm of water and 150 μmol·L^−1^ melatonin sprayed on the leaves), W2CK (irrigated with 300 mm of water and water (instead of melatonin) sprayed on the leaves), W2MT (irrigated with 300 mm of water and 150 μmol·L^−1^ melatonin sprayed on the leaves), W3CK (irrigated with 240 mm of water and water (instead of melatonin) sprayed on the leaves), W3MT (irrigated with 240 mm of water and 150 μmol·L^−1^ melatonin sprayed on the leaves), W4CK (irrigated with 180 mm of water and water (instead of melatonin) sprayed on the leaves), and W4MT (irrigated with 180 mm of water and 150 μmol·L^−1^ melatonin sprayed on the leaves). The results are the mean ± SE of three independent replications. Different letters represent significant differences among treatments (*p* < 0.05).

**Figure 3 ijms-25-13081-f003:**
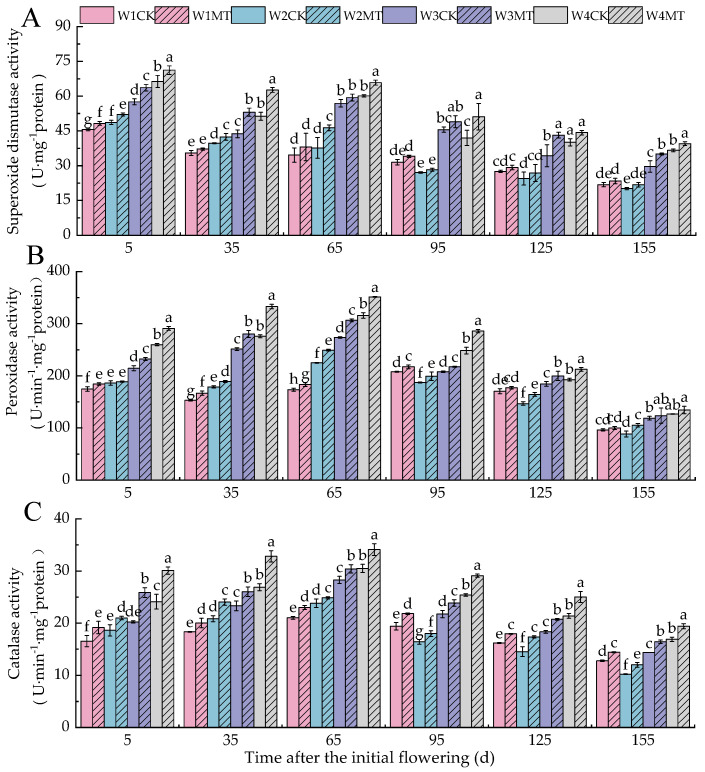
Effect of melatonin on the activity of protective enzymes in grape leaves under varying irrigation amounts (values were determined based on the fresh weight): (**A**) superoxide dismutase (SOD); (**B**) peroxidase (POD); (**C**) catalase (CAT). W1CK (irrigated with 360 mm of water and water (instead of melatonin) sprayed on the leaves), W1MT (irrigated with 360 mm of water and 150 μmol·L^−1^ melatonin sprayed on the leaves), W2CK (irrigated with 300 mm of water and water (instead of melatonin) sprayed on the leaves), W2MT (irrigated with 300 mm of water and 150 μmol·L^−1^ melatonin sprayed on the leaves), W3CK (irrigated with 240 mm of water and water (instead of melatonin) sprayed on the leaves), W3MT (irrigated with 240 mm of water and 150 μmol·L^−1^ melatonin sprayed on the leaves), W4CK (irrigated with 180 mm of water and water (instead of melatonin) sprayed on the leaves), and W4MT (irrigated with 180 mm of water and 150 μmol·L^−1^ melatonin sprayed on the leaves). The results are the mean ± SE of three independent replications. Different letters represent significant differences among treatments (*p* < 0.05).

**Figure 4 ijms-25-13081-f004:**
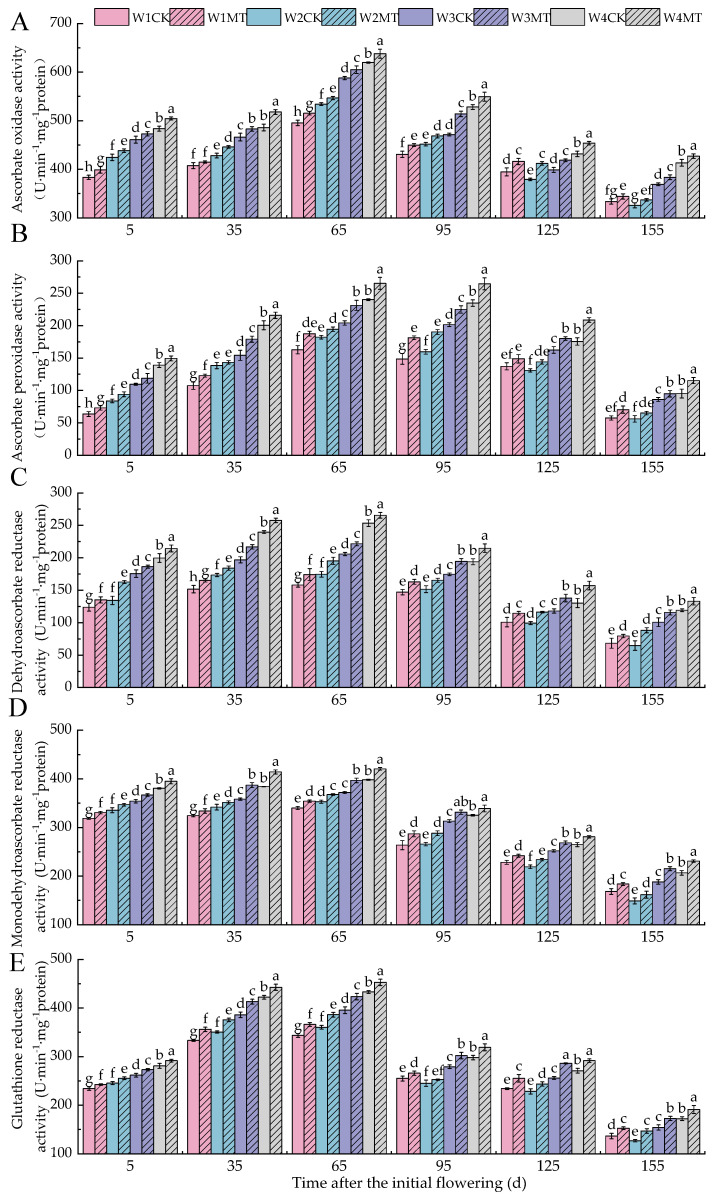
Effects of melatonin on the ascorbate–glutathione cycle enzyme activity in grape leaves under different irrigation amounts (values were determined based on the fresh weight): (**A**) ascorbate oxidase (AAO); (**B**) ascorbate peroxidase (APX); (**C**) dehydroascorbate reductase (DHAR); (**D**) monodehydroascorbate reductase (MDHAR); (**E**) glutathione reductase (GR). W1CK (irrigated with 360 mm of water and water (instead of melatonin) sprayed on the leaves), W1MT (irrigated with 360 mm of water and 150 μmol·L^−1^ melatonin sprayed on the leaves), W2CK (irrigated with 300 mm of water and water (instead of melatonin) sprayed on the leaves), W2MT (irrigated with 300 mm of water and 150 μmol·L^−1^ melatonin sprayed on the leaves), W3CK (irrigated with 240 mm of water and water (instead of melatonin) sprayed on the leaves), W3MT (irrigated with 240 mm of water and 150 μmol·L^−1^ melatonin sprayed on the leaves), W4CK (irrigated with 180 mm of water and water (instead of melatonin) sprayed on the leaves), and W4MT (irrigated with 180 mm of water and 150 μmol·L^−1^ melatonin sprayed on the leaves). The results are the mean ± SE of three independent replications. Different letters represent significant differences among treatments (*p* < 0.05).

**Figure 5 ijms-25-13081-f005:**
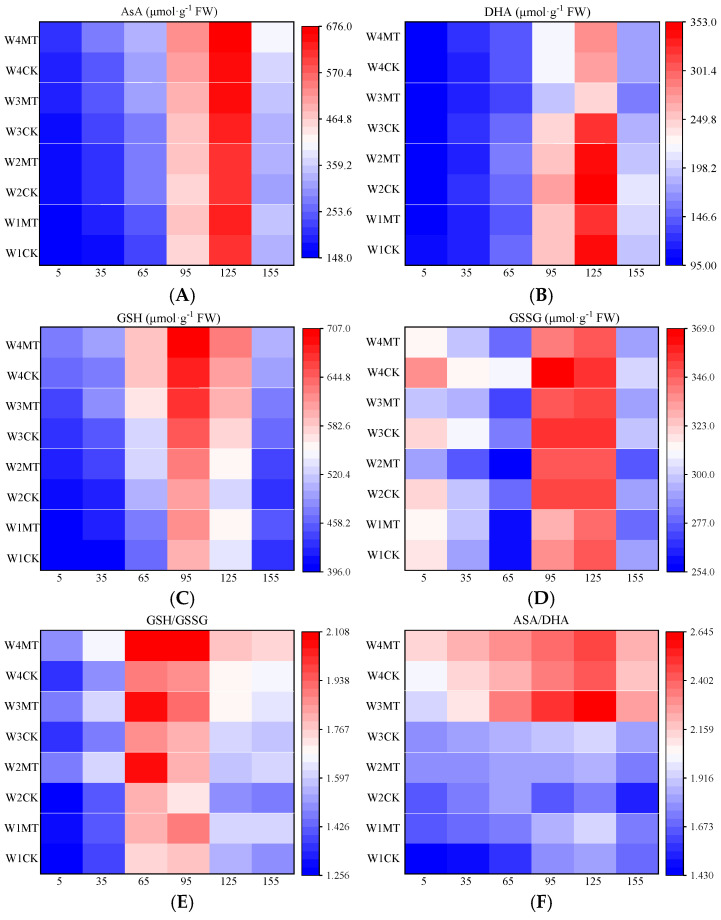
Effects of melatonin on ascorbic acid–glutathione levels in grape leaves under varying irrigation amounts (values were determined based on the fresh weight): (**A**) ascorbic acid (AsA); (**B**) dehydroascorbic acid (DHA); (**C**) reduced glutathione (GSH); (**D**) oxidized glutathione (GSSG); (**E**) ratio of reduced glutathione to oxidized glutathione (GSH/GSSG); (**F**) ratio of ascorbic acid to dehydroascorbic acid (AsA/DHA). W1CK (irrigated with 360 mm of water and water (instead of melatonin) sprayed on the leaves), W1MT (irrigated with 360 mm of water and 150 μmol·L^−1^ melatonin sprayed on the leaves), W2CK (irrigated with 300 mm of water and water (instead of melatonin) sprayed on the leaves), W2MT (irrigated with 300 mm of water and 150 μmol·L^−1^ melatonin sprayed on the leaves), W3CK (irrigated with 240 mm of water and water (instead of melatonin) sprayed on the leaves), W3MT (irrigated with 240 mm of water and 150 μmol·L^−1^ melatonin sprayed on the leaves), W4CK (irrigated with 180 mm of water and water (instead of melatonin) sprayed on the leaves), and W4MT (irrigated with 180 mm of water and 150 μmol·L^−1^ melatonin sprayed on the leaves). The results are the mean ± SE of three independent replications. Different letters represent significant differences among the treatments (*p* < 0.05).

**Figure 6 ijms-25-13081-f006:**
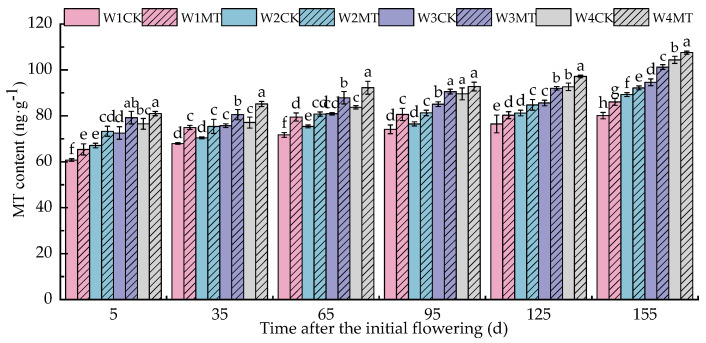
Effect of melatonin on the endogenous MT content in grape leaves under varying irrigation amounts (values were determined based on the fresh weight). W1CK (irrigated with 360 mm of water and water (instead of melatonin) sprayed on the leaves), W1MT (irrigated with 360 mm of water and 150 μmol·L^−1^ melatonin sprayed on the leaves), W2CK (irrigated with 300 mm of water and water (instead of melatonin) sprayed on the leaves), W2MT (irrigated with 300 mm of water and 150 μmol·L^−1^ melatonin sprayed on the leaves), W3CK (irrigated with 240 mm of water and water (instead of melatonin) sprayed on the leaves), W3MT (irrigated with 240 mm of water and 150 μmol·L^−1^ melatonin sprayed on the leaves), W4CK (irrigated with 180 mm of water and water (instead of melatonin) sprayed on the leaves), and W4MT (irrigated with 180 mm of water and 150 μmol·L^−1^ melatonin sprayed on the leaves). The results are the mean ± SE of three independent replications. Different letters represent significant differences among treatments (*p* < 0.05).

**Figure 7 ijms-25-13081-f007:**
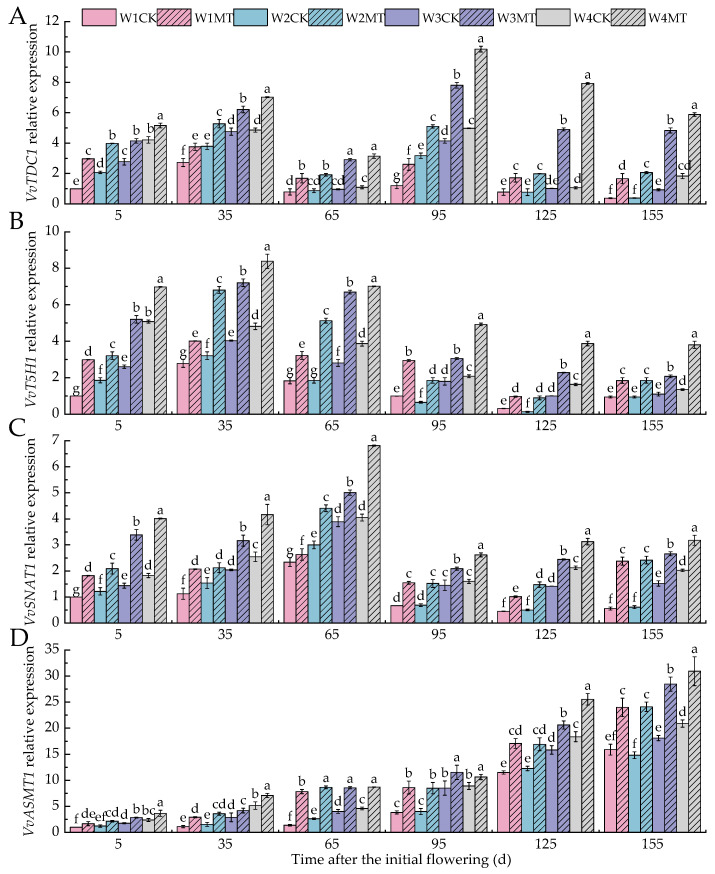
Effects of melatonin on the expression of MT-related genes in grape leaves under varying irrigation amounts: (**A**) *VvTDC1*; (**B**) *VvT5H1;* (**C**) *VvSNAT1;* (**D**) *VvASMT1.* W1CK (irrigated with 360 mm of water and water (instead of melatonin) sprayed on the leaves), W1MT (irrigated with 360 mm of water and 150 μmol·L^−1^ melatonin sprayed on the leaves), W2CK (irrigated with 300 mm of water and water (instead of melatonin) sprayed on the leaves), W2MT (irrigated with 300 mm of water and 150 μmol·L^−1^ melatonin sprayed on the leaves), W3CK (irrigated with 240 mm of water and water (instead of melatonin) sprayed on the leaves), W3MT (irrigated with 240 mm of water and 150 μmol·L^−1^ melatonin sprayed on the leaves), W4CK (irrigated with 180 mm of water and water (instead of melatonin) sprayed on the leaves), and W4MT (irrigated with 180 mm of water and 150 μmol·L^−1^ melatonin sprayed on the leaves). The results are the mean ± SE of three independent replications. Different letters represent significant differences among treatments (*p* < 0.05).

**Figure 8 ijms-25-13081-f008:**
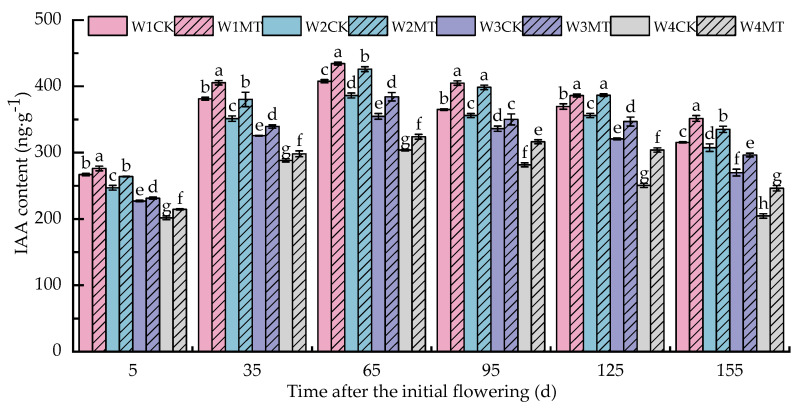
Effect of melatonin on the endogenous IAA content in grape leaves under varying irrigation amounts (values were determined based on the fresh weight). W1CK (irrigated with 360 mm of water and water (instead of melatonin) sprayed on the leaves), W1MT (irrigated with 360 mm of water and 150 μmol·L^−1^ melatonin sprayed on the leaves), W2CK (irrigated with 300 mm of water and water (instead of melatonin) sprayed on the leaves), W2MT (irrigated with 300 mm of water and 150 μmol·L^−1^ melatonin sprayed on the leaves), W3CK (irrigated with 240 mm of water and water (instead of melatonin) sprayed on the leaves), W3MT (irrigated with 240 mm of water and 150 μmol·L^−1^ melatonin sprayed on the leaves), W4CK (irrigated with 180 mm of water and water (instead of melatonin) sprayed on the leaves), and W4MT (irrigated with 180 mm of water and 150 μmol·L^−1^ melatonin sprayed on the leaves). The results are the mean ± SE of three independent replications. Different letters represent significant differences among treatments (*p* < 0.05).

**Figure 9 ijms-25-13081-f009:**
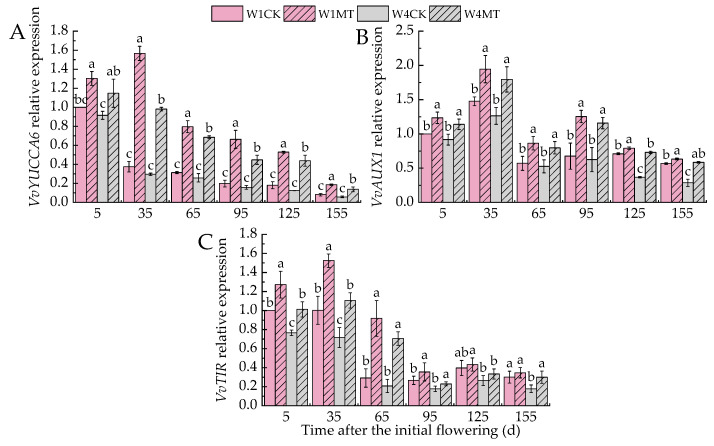
Effects of melatonin on the expression of IAA-related genes in grape leaves under varying irrigation amounts: (**A**) *VvYUCCA6*; (**B**) *VvAUX1;* (**C**) *VvTIR.* W1CK (irrigated with 360 mm of water and water (instead of melatonin) sprayed on the leaves), W1MT (irrigated with 360 mm of water and 150 μmol·L^−1^ melatonin sprayed on the leaves), W2CK (irrigated with 300 mm of water and water (instead of melatonin) sprayed on the leaves), W2MT (irrigated with 300 mm of water and 150 μmol·L^−1^ melatonin sprayed on the leaves), W3CK (irrigated with 240 mm of water and water (instead of melatonin) sprayed on the leaves), W3MT (irrigated with 240 mm of water and 150 μmol·L^−1^ melatonin sprayed on the leaves), W4CK (irrigated with 180 mm of water and water (instead of melatonin) sprayed on the leaves), and W4MT (irrigated with 180 mm of water and 150 μmol·L^−1^ melatonin sprayed on the leaves). The results are the mean ± SE of three independent replications. Different letters represent significant differences among treatments (*p* < 0.05).

**Figure 10 ijms-25-13081-f010:**
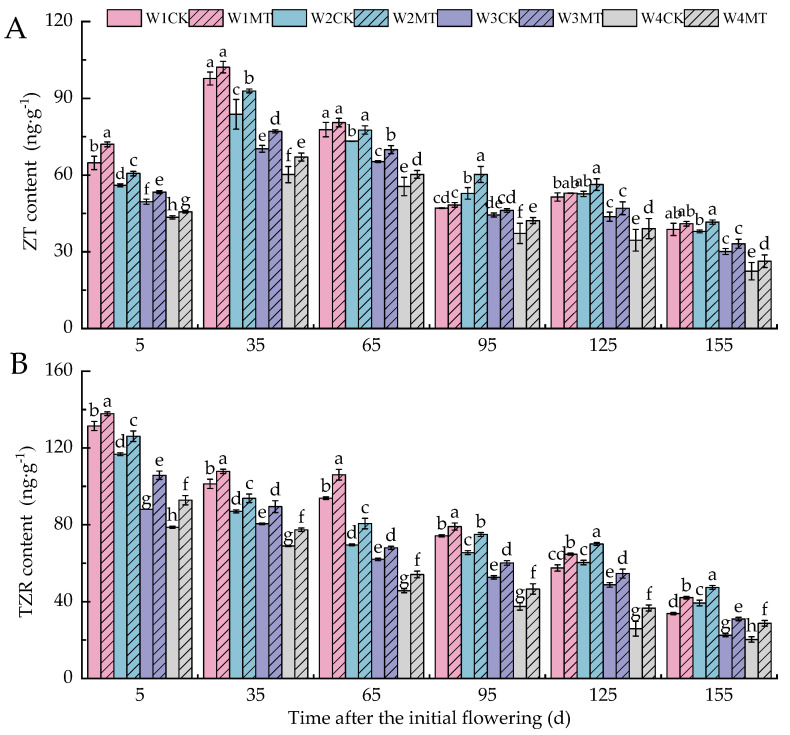
Effects of melatonin on the contents of endogenous ZT and TZR in grape leaves under varying irrigation amounts (values were determined based on the fresh weight): (**A**) zeatin (ZT); (**B**) trans-zeatin (TZR). W1CK (irrigated with 360 mm of water and water (instead of melatonin) sprayed on the leaves), W1MT (irrigated with 360 mm of water and150 μmol·L^−1^ melatonin sprayed on the leaves), W2CK (irrigated with 300 mm of water and water (instead of melatonin) sprayed on the leaves), W2MT (irrigated with 300 mm of water and 150 μmol·L^−1^ melatonin sprayed on the leaves), W3CK (irrigated with 240 mm of water and water (instead of melatonin) sprayed on the leaves), W3MT (irrigated with 240 mm of water and 150 μmol·L^−1^ melatonin sprayed on the leaves), W4CK (irrigated with 180 mm of water and water (instead of melatonin) sprayed on the leaves), and W4MT (irrigated with 180 mm of water and 150 μmol·L^−1^ melatonin sprayed on the leaves). The results are the mean ± SE of three independent replications. Different letters represent significant differences among treatments (*p* < 0.05).

**Figure 11 ijms-25-13081-f011:**
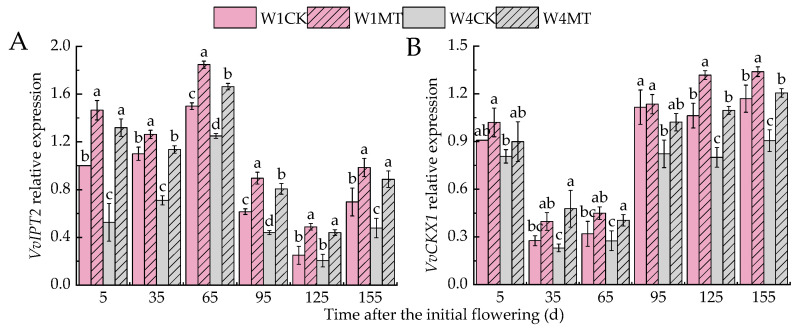
Effects of melatonin on the expression of ZT- and TZR-related genes in grape leaves under varying irrigation amounts: (**A**) *VvIPT2*; (**B**) *VvCKX1.* W1CK (irrigated with 360 mm of water and water (instead of melatonin) sprayed on the leaves), W1MT (irrigated with 360 mm of water and 150 μmol·L^−1^ melatonin sprayed on the leaves), W2CK (irrigated with 300 mm of water and water (instead of melatonin) sprayed on the leaves), W2MT (irrigated with 300 mm of water and 150 μmol·L^−1^ melatonin sprayed on the leaves), W3CK (irrigated with 240 mm of water and water (instead of melatonin) sprayed on the leaves), W3MT (irrigated with 240 mm of water and 150 μmol·L^−1^ melatonin sprayed on the leaves), W4CK (irrigated with 180 mm of water and water (instead of melatonin) sprayed on the leaves), and W4MT (irrigated with 180 mm of water and 150 μmol·L^−1^ melatonin sprayed on the leaves). The results are the mean ± SE of three independent replications. Different letters represent significant differences among treatments (*p* < 0.05).

**Figure 12 ijms-25-13081-f012:**
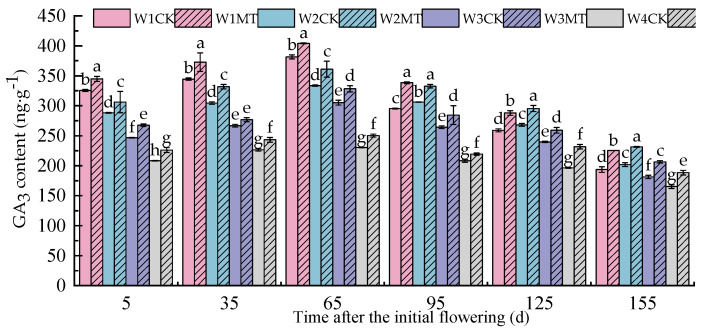
Effect of melatonin on the content of endogenous GA_3_ in grape leaves under varying irrigation amounts (values were determined based on the fresh weight). W1CK (irrigated with 360 mm of water and water (instead of melatonin) sprayed on the leaves), W1MT (irrigated with 360 mm of water and 150 μmol·L^−1^ melatonin sprayed on the leaves), W2CK (irrigated with 300 mm of water and water (instead of melatonin) sprayed on the leaves), W2MT (irrigated with 300 mm of water and 150 μmol·L^−1^ melatonin sprayed on the leaves), W3CK (irrigated with 240 mm of water and water (instead of melatonin) sprayed on the leaves), W3MT (irrigated with 240 mm of water and150 μmol·L^−1^ melatonin sprayed on the leaves), W4CK (irrigated with 180 mm of water and water (instead of melatonin) sprayed on the leaves), and W4MT (irrigated with 180 mm of water and 150 μmol·L^−1^ melatonin sprayed on the leaves). The results are the mean ± SE of three independent replications. Different letters represent significant differences among treatments (*p* < 0.05).

**Figure 13 ijms-25-13081-f013:**
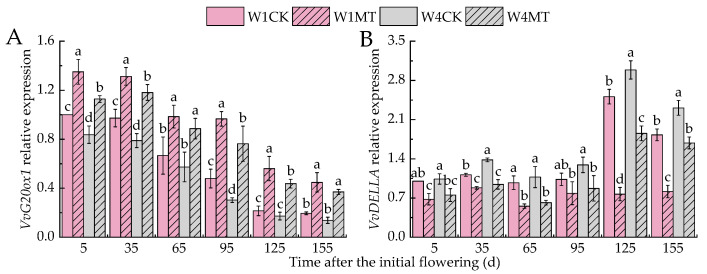
Effects of melatonin on the expression of GA_3_-related genes in grape leaves under varying irrigation amounts: (**A**)*VvG20ox1*; (**B**)*VvDELLA*. W1CK (irrigated with 360 mm of water and water (instead of melatonin) sprayed on the leaves), W1MT (irrigated with 360 mm of water and 150 μmol·L^−1^ melatonin sprayed on the leaves), W2CK (irrigated with 300 mm of water and water (instead of melatonin) sprayed on the leaves), W2MT (irrigated with 300 mm of water and 150 μmol·L^−1^ melatonin sprayed on the leaves), W3CK (irrigated with 240 mm of water and water (instead of melatonin) sprayed on the leaves), W3MT (irrigated with 240 mm of water and 150 μmol·L^−1^ melatonin sprayed on the leaves), W4CK (irrigated with 180 mm of water and water (instead of melatonin) sprayed on the leaves), and W4MT (irrigated with 180 mm of water and 150 μmol·L^−1^ melatonin sprayed on the leaves). The results are the mean ± SE of three independent replications. Different letters represent significant differences among treatments (*p* < 0.05).

**Figure 14 ijms-25-13081-f014:**
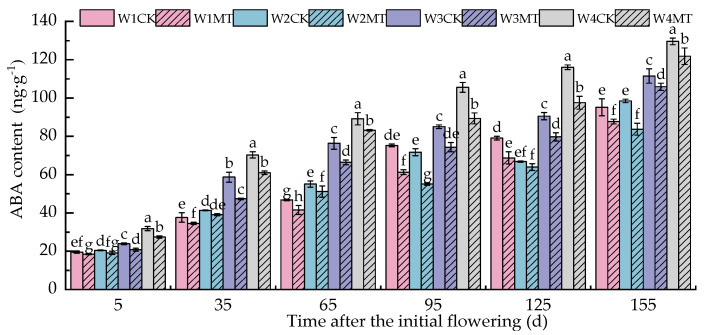
Effect of melatonin on the endogenous ABA content in grape leaves under varying irrigation amounts (values were determined based on the fresh weight). W1CK (irrigated with 360 mm of water and water (instead of melatonin) sprayed on the leaves), W1MT (irrigated with 360 mm of water and150 μmol·L^−1^ melatonin sprayed on the leaves), W2CK (irrigated with 300 mm of water and water (instead of melatonin) sprayed on the leaves), W2MT (irrigated with 300 mm of water and 150 μmol·L^−1^ melatonin sprayed on the leaves), W3CK (irrigated with 240 mm of water and water (instead of melatonin) sprayed on the leaves), W3MT (irrigated with 240 mm of water and 150 μmol·L^−1^ melatonin sprayed on the leaves), W4CK (irrigated with 180 mm of water and water (instead of melatonin) sprayed on the leaves), and W4MT (irrigated with 180 mm of water and 150 μmol·L^−1^ melatonin sprayed on the leaves). The results are the mean ± SE of three independent replications. Different letters represent significant differences among treatments (*p* < 0.05).

**Figure 15 ijms-25-13081-f015:**
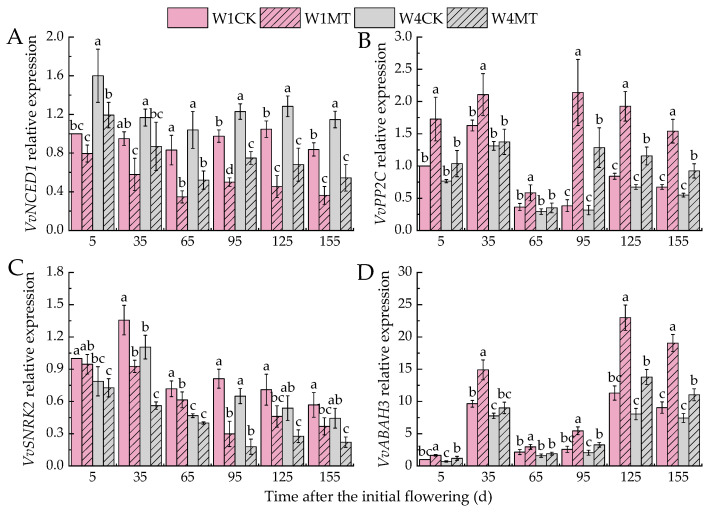
Effects of melatonin on the expression of ABA-related genes in grape leaves under varying irrigation amounts: (**A**) *VvNCED1*; (**B**) *VvPP2C*; (**C**) *VvSNRK2*; (**D**) *VvABAH3*. W1CK (irrigated with 360 mm of water and water (instead of melatonin) sprayed on the leaves), W1MT (irrigated with 360 mm of water and 150 μmol·L^−1^ melatonin sprayed on the leaves), W2CK (irrigated with 300 mm of water and water (instead of melatonin) sprayed on the leaves), W2MT (irrigated with 300 mm of water and 150 μmol·L^−1^ melatonin sprayed on the leaves), W3CK (irrigated with 240 mm of water and water (instead of melatonin) sprayed on the leaves), W3MT (irrigated with 240 mm of water and 150 μmol·L^−1^ melatonin sprayed on the leaves), W4CK (irrigated with 180 mm of water and water (instead of melatonin) sprayed on the leaves), and W4MT (irrigated with 180 mm of water and 150 μmol·L^−1^ melatonin sprayed on the leaves). The results are the mean ± SE of three independent replications. Different letters represent significant differences among treatments (*p* < 0.05).

**Figure 16 ijms-25-13081-f016:**
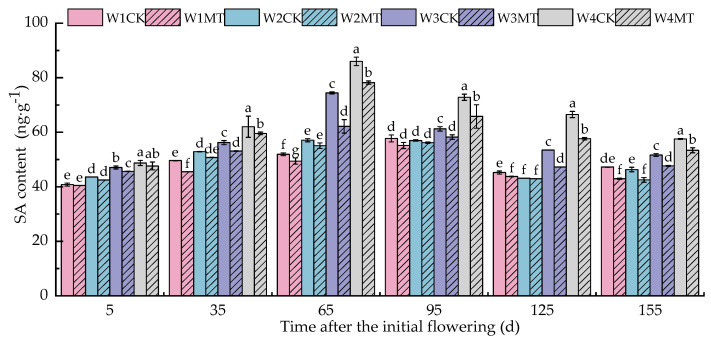
Effect of melatonin on endogenous SA content in grape leaves under varying irrigation amounts (values were determined based on the fresh weight). W1CK (irrigated with 360 mm of water and water (instead of melatonin) sprayed on the leaves), W1MT (irrigated with 360 mm of water and 150 μmol·L^−1^ melatonin sprayed on the leaves), W2CK (irrigated with 300 mm of water and water (instead of melatonin) sprayed on the leaves), W2MT (irrigated with 300 mm of water and 150 μmol·L^−1^ melatonin sprayed on the leaves), W3CK (irrigated with 240 mm of water and water (instead of melatonin) sprayed on the leaves), W3MT (irrigated with 240 mm of water and 150 μmol·L^−1^ melatonin sprayed on the leaves), W4CK (irrigated with 180 mm of water and water (instead of melatonin) sprayed on the leaves), and W4MT (irrigated with 180 mm of water and 150 μmol·L^−1^ melatonin sprayed on the leaves). The results are the mean ± SE of three independent replications. Different letters represent significant differences among treatments (*p* < 0.05).

**Table 1 ijms-25-13081-t001:** Effects of the melatonin on the soluble solids and titratable acid contents in grape berries under varying irrigation amounts.

Treatment	Soluble Solids (%)	Titratable Acid (%)	Solid Acid Ratio	Firmness (kg·cm^−2^)
W1CK	16.83 ± 0.23 e	0.35 ± 0.01 a	48.6 ± 1.98 h	0.61 ± 0.02 f
W1MT	17.97 ± 0.31 d	0.34 ± 0.01 a	53.47 ± 1.46 g	0.72 ± 0.02 de
W2CK	20.5 ± 0.30 b	0.29 ± 0.01 cd	69.74 ± 2.48 d	0.75 ± 0.04 cd
W2MT	21.4 ± 0.11 a	0.28 ± 0.01 de	75.1 ± 1.01 c	0.79 ± 0.03 bc
W3CK	21.53 ± 0.35 a	0.27 ± 0.01 e	78.57 ± 1.84 b	0.81 ± 0.01 b
W3MT	21.63 ± 0.76 a	0.25 ± 0.01 f	85.94 ± 1.63 a	0.99 ± 0.03 a
W4CK	18.27 ± 0.15 d	0.32 ± 0.02 b	57.38 ± 0.14 f	0.7 ± 0.02 e
W4MT	19.13 ± 0.21 c	0.3 ± 0.01 cd	63.36 ± 1.66 e	0.75 ± 0.04 cd

The values were determined based on the fresh weight. W1CK (irrigated with 360 mm of water and water (instead of melatonin) sprayed on the leaves), W1MT (irrigated with 360 mm of water and 150 μmol·L^−1^ melatonin sprayed on the leaves), W2CK (irrigated with 300 mm of water and water (instead of melatonin) sprayed on the leaves), W2MT (irrigated with 300 mm of water and 150 μmol·L^−1^ melatonin sprayed on the leaves), W3CK (irrigated with 240 mm of water and water (instead of melatonin) sprayed on the leaves), W3MT (irrigated with 240 mm of water and 150 μmol·L^−1^ melatonin sprayed on the leaves), W4CK (irrigated with 180 mm of water and water (instead of melatonin) sprayed on the leaves), and W4MT (irrigated with 180 mm of water and 150 μmol·L^−1^ melatonin sprayed on the leaves). The results are the mean ± SE of three independent replications. Different letters represent significant differences among treatments (*p* < 0.05).

**Table 2 ijms-25-13081-t002:** Effects of the melatonin on the soluble sugar and sugar fractions in grape berries under varying irrigation amounts.

Treatment	Total Soluble Sugar (mg·g^−1^)	Glucose (mg·g^−1^)	Fructose (mg·g^−1^)	Sucrose (mg·g^−1^)
W1CK	72.46 ± 2.46 d	28.91 ± 0.26 e	24.97 ± 1.21 f	2.50 ± 0.02 c
W1MT	74.88 ± 2.95 d	29.87 ± 0.50 e	26.14 ± 0.19 f	2.60 ± 0.05 c
W2CK	78.32 ± 5.13 cd	33.06 ± 0.60 d	28.83 ± 1.47 e	2.52 ± 0.03 c
W2MT	82.40 ± 3.62 bc	34.43 ± 0.57 cd	32.55 ± 0.84 d	2.81 ± 0.12 b
W3CK	88.10 ± 3.68 ab	35.48 ± 0.67 bc	33.93 ± 0.99 cd	2.80 ± 0.11 b
W3MT	92.07 ± 4.38 a	36.71 ± 0.87 bc	34.81 ± 0.40 bc	2.91 ± 0.14 ab
W4CK	93.31 ± 3.51 a	40.11 ± 1.06 a	36.05 ± 0.70 ab	3.03 ± 0.02 a
W4MT	95.07 ± 4.46 a	40.69 ± 1.66 a	37.57 ± 1.07 a	3.05 ± 0.06 a

The values were determined based on the fresh weight. W1CK (irrigated with 360 mm of water and water (instead of melatonin) sprayed on the leaves), W1MT (irrigated with 360 mm of water and 150 μmol·L^−1^ melatonin sprayed on the leaves), W2CK (irrigated with 300 mm of water and water (instead of melatonin) sprayed on the leaves), W2MT (irrigated with 300 mm of water and 150 μmol·L^−1^ melatonin sprayed on the leaves), W3CK (irrigated with 240 mm of water and water (instead of melatonin) sprayed on the leaves), W3MT (irrigated with 240 mm of water and 150 μmol·L^−1^ melatonin sprayed on the leaves), W4CK (irrigated with 180 mm of water and water (instead of melatonin) sprayed on the leaves), and W4MT (irrigated with 180 mm of water and 150 μmol·L^−1^ melatonin sprayed on the leaves). The results are the mean ± SE of three independent replications. Different letters represent significant differences among treatments (*p* < 0.05).

**Table 3 ijms-25-13081-t003:** Effects of the melatonin on the phenolic compounds and vitamin C in grape berries under varying irrigation amounts.

Treatment	Total Phenols (mg·g^−1^)	Tannin (mg·g^−1^)	Total Flavonoids (mg·g^−1^)	Vitamin C (mg·100 g^−1^)
W1CK	4.31 ± 0.23 g	2.37 ± 0.03 f	1.72 ± 0.02 e	8.02 ± 0.21 d
W1MT	4.70 ± 0.23 f	2.32 ± 0.08 f	1.91 ± 0.01 d	9.28 ± 0.25 cd
W2CK	5.47 ± 0.06 d	2.49 ± 0.01 e	1.94 ± 0.04 d	11.63 ± 0.10 a
W2MT	5.75 ± 0.09 c	2.39 ± 0.01 f	2.31 ± 0.01 c	11.88 ± 0.15 a
W3CK	6.03 ± 0.17 b	2.87 ± 0.03 c	2.42 ± 0.07 b	10.78 ± 0.25 ab
W3MT	6.38 ± 0.13 a	2.59 ± 0.01 d	2.64 ± 0.04 a	11.48 ± 0.05 a
W4CK	4.95 ± 0.02 ef	3.37 ± 0.07 a	2.35 ± 0.05 c	10.03 ± 2.01 bc
W4MT	5.21 ± 0.14 de	3.15 ± 0.05 b	2.43 ± 0.03 b	10.97 ± 0.44 ab

The values were determined based on the fresh weight. W1CK (irrigated with 360 mm of water and water (instead of melatonin) sprayed on the leaves), W1MT (irrigated with 360 mm of water and150 μmol·L^−1^ melatonin sprayed on the leaves), W2CK (irrigated with 300 mm of water and water (instead of melatonin) sprayed on the leaves), W2MT (irrigated with 300 mm of water and 150 μmol·L^−1^ melatonin sprayed on the leaves), W3CK (irrigated with 240 mm of water and water (instead of melatonin) sprayed on the leaves), W3MT (irrigated with 240 mm of water and 150 μmol·L^−1^ melatonin sprayed on the leaves), W4CK (irrigated with 180 mm of water and water (instead of melatonin) sprayed on the leaves), and W4MT (irrigated with 180 mm of water and 150 μmol·L^−1^ melatonin sprayed on the leaves). The results are the mean ± SE of three independent replications. Different letters represent significant differences among treatments (*p* < 0.05).

**Table 4 ijms-25-13081-t004:** Effects of the melatonin on the yield of grape berries under the varying irrigation amounts.

Treatment	Single Fruit Weight (g)	Vertical Diameter (mm)	Transverse Diameter (mm)	Index of Fruit Figure
W1CK	10.94 ± 0.24 b	28.32 ± 0.22 c	26.37 ± 0.23 d	1.074 ± 0.011 c
W1MT	12.46 ± 0.19 a	29.68 ± 0.19 b	27.7 ± 0.07 b	1.071 ± 0.010 c
W2CK	11.23 ± 0.07 b	29.57 ± 0.17 b	27.01 ± 0.10 c	1.095 ± 0.007 bc
W2MT	12.72 ± 0.16 a	30.77 ± 0.27 a	28.26 ± 0.04 a	1.089 ± 0.011 bc
W3CK	10.27 ± 0.24 c	27.3 ± 0.41 d	24.37 ± 0.29 f	1.12 ± 0.030 b
W3MT	11.09 ± 0.50 b	28.56 ± 0.46 c	25.52 ± 0.32 e	1.119 ± 0.025 b
W4CK	9.25 ± 0.09 d	26.1 ± 0.17 e	22.28 ± 0.59 h	1.172 ± 0.025 a
W4MT	9.92 ± 0.16 c	27.05 ± 0.44 d	23.37 ± 0.31 g	1.158 ± 0.021 a

The values were determined based on the fresh weight. W1CK (irrigated with 360 mm of water and water (instead of melatonin) sprayed on the leaves), W1MT (irrigated with 360 mm of water and 150 μmol·L^−1^ melatonin sprayed on the leaves), W2CK (irrigated with 300 mm of water and water (instead of melatonin) sprayed on the leaves), W2MT (irrigated with 300 mm of water and 150 μmol·L^−1^ melatonin sprayed on the leaves), W3CK (irrigated with 240 mm of water and water (instead of melatonin) sprayed on the leaves), W3MT (irrigated with 240 mm of water and 150 μmol·L^−1^ melatonin sprayed on the leaves), W4CK (irrigated with 180 mm of water and water (instead of melatonin) sprayed on the leaves), and W4MT (irrigated with 180 mm of water and 150 μmol·L^−1^ melatonin sprayed on the leaves). The results are the mean ± SE of three independent replications. Different letters represent significant differences among treatments (*p* < 0.05).

**Table 5 ijms-25-13081-t005:** The principal component score table of the MT treatment on grapes under varying irrigation amounts.

Treatment	Principal Component Score				Comprehensive Score (*F*)	Ranking
	FAC1	FAC2	FAC3	FAC4		
W1CK	−1.21	−1.15	0.39	−0.69	−106.01	8
W1MT	−0.98	−0.22	1.18	−0.30	−67.76	7
W2CK	−0.80	0.16	−1.37	0.60	−63.24	6
W2MT	−0.53	1.15	0.18	1.37	−14.89	5
W3CK	0.36	0.25	−1.38	−0.37	19.76	4
W3MT	0.69	1.48	0.44	−1.61	73.22	2
W4CK	1.16	−1.39	−0.52	−0.20	57.94	3
W4MT	1.31	−0.28	1.08	1.19	100.98	1
Eigen value	29.13	6.51	2.71	1.07		
Variance contribution (%)	72.83	16.27	6.78	2.67		
Cumulative variance proportion (%)	72.83	89.11	95.89	98.55		

**Table 6 ijms-25-13081-t006:** Real-time fluorescence quantitative *q*RT–PCR primer sequence.

Primer Name	NCBI Login Number	Primer Sequence (5′-3′)
*VvTDC1*	*XM_010654123.2*	F: ATGGAGAGTGGACTGAGGCCCA R: TGATTAGGTGCGGAATCTGGCA
*VvT5H1*	*XM_002276522.4*	F: GCCATCATTGGCAACCTTCATC R: GCCAGATCATGGGTCTTCATCACT
*VvSNAT1*	*XM_002266325.4*	F: CGCCCTCCTCTCCACTTCTCAG R: GCTTTCTTCTTGCTCTCCCAACCC
*VvASMT1*	*XM_002278056.3*	F: AGGTATGTGAACCGGCTCATGC R: TCGAGCATTGCCAGAACGAAG
*VvYUCCA6*	*XM_002267965.3*	F: TGGACAGGAGGTTCGATGGACTAAG R: GTTCCGCATTCTCACCAGTAGCC
*VvAUX1*	*XM_002279183.4*	F: TTCGTGTGGGAGAAGGTGATAGGG R: TCGGGATAACAACTGGGAGTCTGG
*VvTIR*	*XM_002269091.4*	F: ACTACATCGCCTTTCCCTCTCTGG R: TTTCTAGCTTCTTGGCATGGGTTCC
*VvIPT2*	*XM_019222408.1*	F: CCAGGTTTCCCGCAGAGATTGTG R: CTGCTCCTCCTCTGTGATCTTGTTG
*VvCKX1*	*XM_002284524.3*	F: ACCTTCCATCGGCAATTCTACATCC R: CCTCTAGCGGCAATGGTTAGTTCTG
*VvG20ox1*	*NM_001319281.1*	F: GCCACCTGAGCTTCTTGTTCCTC R: AAGACGAGCCGCATTTGAGATGG
*VvDELLA*	*NM_001397856.1*	F: TACAGGGTGGAGGAGAACAATGGG R: TCAGTTGGAGGCAGGTGTGGAG
*VvNCED1*	*XM_019216859.1*	F: AATGCTACTGACGCTTCTGGTATGC R: CAGGAGCCAATCACCACGACTTC
*VvPP2C*	*XM_002283400.3*	F: CTTGAGGAGCGAGAACGTGTTACC R: ACAGCAGGCTTCAGATCATCATCAC
*VvSNRK2A*	*XM_002269185.3*	F: GTGGCTAGGCTTATGAGGAACAAGG R: TATGTTTGGATGCCGAAGGGAACG
*VvABAH3*	*XM_010650812.2*	F: GCAGGCGAAGTGGAAAGGAGTAC R: CAGCCGCAGTGGTGGTTTGTC
*VvPAL*	*NM_001397918.1*	F: CACCAGGGGAGGATTTTGACAAGG R: GAGCACCGTTCCAAGCACTGAG
*VvNPR1*	*XM_002281439.4*	F: GACTTCTTCACAGACGCCAGGATC R: ACTTCGCCTCCTTCTCCTTCCTC
*Actin*	*XM_002263109.3*	F: TTCTCGTTGAGGGCTATTCCA R: CCACAGACTTCATCGGTGACA

## Data Availability

All data are presented in the manuscript.

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
