# Peer review of "Melatonin-Mediated Modulation of Grapevine Resistance Physiology, Endogenous Hormonal Dynamics, and Fruit Quality Under Varying Irrigation Amounts"

_ijms, 2024, doi:10.3390/ijms252313081_

Round 1

Reviewer 1 Report

Comments and Suggestions for Authors

Abstract: too long and too didactic, it should be shortened and avoid stating all parameters and genes analyzed, better to select the most relevant ones. Also, abbreviations/acronyms should be explained at their first introduction.

Keywords: should be replaced because they are unnecessary as they are already included in the title.

Introduction: please shorten the last paragraph, which is too detailed. Also, if the purpose of the stated work is to improve grape quality and yield (lines 83-85), the yields resulting from different treatments are missing from the results.

Results:

-          Treatment abbreviations need to be replaced in a way that makes them easy to understand; for this, at the very least, the “W” should be removed.

-          Need to modify Figure 1 (and subsequent figures) to provide a clear representation of treatments and data over time: a) inserting a small space after each histogram pair (+ or - Met), with a softer color for CK, since a comparison over time between histogram pairs is shown; b) indicate, e.g., with arrows the times of melatonin treatments (days 0, 30, 60 and 90 days post-anthesis); c) indicate whether units refer to fresh weight or dry weight although results by dry weight are more appropriate in relation to reduced water availability; d) if “Time after the initial flowering (days)” is indicated at the base of the axis, only numbers should be given; e) In the text (lines 91-97) regarding Figure 1 and in the figure legend, the correspondence between the abbreviations and the level of water administered (decreasing from W1 to W4) is not explained, and only in M&Ms comes the confirmation that W1 corresponds to 360 mm of water, W2 to 300 mm, W3 to 240 mm, and W4 to180 mm; therefore, the abbreviations of the treatments should also be explained when they are first introduced and it should be kept in mind that generally the figure axes represent increasing values from bottom to top and from left to right.

-          Figures 3 and 4: a) enzyme units (U is defined as the amount of enzyme required to convert 1 µmol of substrate to product in 1 min) should be expressed with respect to the soluble proteins present in the analyzed sample; b) representation with histograms is certainly easier to understand than the actual lines.

-          Figure 5: appropriate to separate the different treatment pairs with a line to facilitate understanding.

-          Table 3: in this case it is appropriate to express data by fresh weight, but berries yield should be added.

-          Figure 17 should be removed or included in Supplementary Materials, it is in the current form unreadable.

Materials and Methods:

- How much rainfall during the growing season must be indicated.

- Grain selection criteria, including freedom from pests and diseases (line 682), must be explained because in the case of different levels of discard between theses, the results are unreliable.

- Regarding the catalase assay the cited reference does not seem appropriate; please consult Aebi, H. (1974) Catalase. In: Bergmeyer, H.U., Ed., Methods of Enzymatic Analysis, http://dx.doi.org/10.1016/b978-0-12-091302-2.50032-3, or Aebi H (1984) Catalase in vitro, DOI: 10.1016/s0076-6879(84)05016-3; in addition, the definition of the unit of measurement (lines 705-706) is incorrect (see commentary for Figures 3 and 4).

The conclusions are very weak, not sufficiently highlighting the importance of the results presented and especially the last lines “Furthermore, MT was beneficial for increasing the content of total soluble sugars, glucose, fructose and sucrose in the fruit, thereby improving the flavor quality of the fruit. Principal component analysis confirmed that the 150 μmol·L-1 MT treatment had the best effect under 180 mm of irrigation (W4), which is suitable for high-quality and efficient grape production and cultivation“ (lines 776-778). Therefore this part should be corrected in that: a) no flavor analysis is presented; b) melatonin treatments are repeated over time; c) the claim that melatonin has the greatest effect at with 180 mm irrigation should be emphasized in physiological terms but moderated considering that yield was limited (add yield data) and in practice it is not conceivable to grow vines with 280 mm of water per year (100 from rainfall and 180 from irrigation) when the “annual evaporation is approximately 2020 mm” (line 658).

Author Response

Comments 1: Abstract: too long and too didactic, it should be shortened and avoid stating all parameters and genes analyzed, better to select the most relevant ones. Also, abbreviations / acronyms should be explained at their first introduction.

Response 1: Agreeing with the reviewer, we have rewritten the abstract and highlighted it with a yellow background; the detailed information can be found in lines 11- 24 of the revised version.

Comments 2: Keywords: should be replaced because they are unnecessary as they are already included in the title.

Response 2: Agreeing with the reviewer, we have replaced the keywords and highlighted them with a yellow background; the detailed information can be found in lines 25-26 of the revised version.

Comments 3: Introduction: please shorten the last paragraph, which is too detailed. Also, if the purpose of the stated work is to improve grape quality and yield (lines 83-85), the yields resulting from different treatments are missing from the results.

Response 3: Agreeing with the reviewer, a) We have revised the last paragraph of the introduction and highlighted it with a yellow background; the detailed information can be found in lines 75- 82of the revised version. b) We sincerely apologize that due to experimental design and time constraints, we were unable to measure the fruit yield parameters. However, we have measured the single fruit weight, longitudinal and transverse diameter, and fruit shape index. This additional content has been included in the Results section (Figure 5) with a yellow background highlight, and the detailed information can be found in lines 543-565 of the revised version.

Comments 4: Results: (1) Treatment abbreviations need to be replaced in a way that makes them easy to understand; for this, at the very least, the “W” should be removed. (2) Need to modify Figure 1 (and subsequent figures). a) inserting a small space after each histogram pair (+ or - Met), with a softer color for CK, since a comparison over time between histogram pairs is shown; b) indicate, eg, with arrows the times of melatonin treatments (days 0, 30, 60 and 90 days post-anthesis); c) indicate whether units refer to fresh weight or dry weight although results by dry weight are more appropriate in relation to reduced water availability; d) if “Time after the initial flowering (days)” is indicated at the base of the axis, only numbers should be given; e) In the text (lines 91-97) regarding Figure 1 and in the figure legend, the correspondence between the abbreviations and the level of water administered (decreasing from W1 to W4) is not explained, and only in M&Ms comes the confirmation that W1 corresponds to 360 mm of water, W2 to 300 mm, W3 to 240 mm, and W4 to180 mm; therefore, the abbreviations of the treatments should also be explained when they are first introduced and it should be kept in mind that generally the figure axes represent increasing values from bottom to top and from left to right.

Response 4: (1) We respectfully disagree with the reviewer on the point about "Treatment abbreviations" or removing "W". We believe it is easily understood because "W" stands for water, representing the amount of irrigation; MT is the abbreviation for Melatonin, indicating that Melatonin (150 μmol·L⁻¹) was applied foliarly; CK (control) was sprayed with water instead of Melatonin. For example: W1CK (360mm of irrigation amount and application of water), W1MT (360mm of irrigation amount and application of 150 μmol·L⁻¹ melatonin), etc. (2) We sincerely apologize for not changing the color of CK. Although it is reasonable to choose a softer color for CK, there are four MT treatments and correspondingly four CK treatments in this figure. If we change the CK color, it would result in eight different colors in the figure, which would be visually cluttered and unclear. c) refers to fresh weight, which we have supplemented in Figure 1 (and subsequent figures), and added yield parameters to the results section with a yellow background for emphasis, detailed information can be found on lines 109 of the revised version. d) We have made the modifications, see details in Figures 1 to 16 of the revised manuscript. e) The meanings of the treatments WICK, W1MT, W2CK, W2MT, W3CK, W3MT, W4CK, W4MT are as follows: W1CK (360mm of irrigation amount and application of water), W1MT (360mm of irrigation amount and application of 150 μmol·L⁻¹ melatonin), W2CK (240mm of irrigation amount and application of water), W2MT (240mm of irrigation amount and application of 150 μmol·L⁻¹ melatonin), W3CK (180mm of irrigation amount and application of water), W3MT (180mm of irrigation amount and application of 150 μmol·L⁻¹ melatonin), W4CK (120mm of irrigation amount and application of water), W4MT (120mm of irrigation amount and application of 150 μmol·L⁻¹ melatonin). We have added this explanation under the titles of Figures 1 to 16 and table 2 to 5 in the revised manuscript, highlighted with a yellow background.

Comments 5:  Figures 3 and 4: a) enzyme units (U is defined as the amount of enzyme required to convert 1 µmol of substrate to product in 1 min) should be expressed with respect to the soluble proteins present in the analyzed sample; b) representation with histograms is certainly easier to understand than the actual lines.

Response 5: Agreeing with the reviewer, a) We have changed the units of enzymes in Figures 3 and 4. b) and have also converted the line charts to bar graphs. refer to lines 165 and 199 of the revised manuscript respectively.

Comments 6:  Figure 5: appropriate to separate the different treatment pairs with a line to facilitate understanding.

Response 6: Agreeing with the reviewer, we have modified Figure 5 to include a white line separating each treatment group.

Comments 7: Table 3: in this case it is appropriate to express data by fresh weight, but berries yield should be added.

Response 7: We sincerely apologize that due to experimental design and time constraints, we were unable to measure the fruit yield parameters. However, we have measured the single fruit weight, longitudinal and transverse diameter, and fruit shape index. This additional content has been included in the Results section (Figure 5) with a yellow background highlight, and the detailed information can be found in lines 543-565 of the revised version.

Comments 8:  Figure 17 should be removed or included in Supplementary materials, it is in the current form unreadable.

Response 8: Agreeing with the reviewer, we have removed Figure 17.

Comments 9: How much rainfall during the growing season must be indicated.

Response 9:  Agreeing with the reviewer, upon reviewing the literature, The precipitation during the growing season is approximately 150 to 250 mm. This information has been supplemented in the revised manuscript, highlighted with a yellow background, on line 768-769.

Comments 10: Grain selection criteria, including freedom from pests and diseases (line 682), must be explained because in the case of different levels of discard between theses, the results are unreliable.

Response 10: Agreeing with the reviewer, during the fruit sampling process, we did not select ears that were free of pests and diseases or mechanical damage. Instead, we randomly picked one ear from each grapevine. There was a mistake in the manuscript, which has been corrected and highlighted with a yellow background; the detailed information can be found in lines 791-793 of the revised version.

Comments 11: Regarding the catalase assay the cited reference does not seem appropriate; please consult Aebi, H. (1974) Catalase. In: Bergmeyer, H.U., Ed., Methods of Enzymatic Analysis, http://dx.doi.org/10.1016/b978-0-12-091302-2.50032-3, or Aebi H (1984) Catalase in vitro, DOI: 10.1016/s0076-6879(84)05016-3; in addition, the definition of the unit of measurement (lines 705-706) is incorrect (see commentary for Figures 3 and 4).

Response 11: Agreeing with the reviewer, a) We have replaced the reference cited for the determination of catalase activity, and highlighted it with a yellow background in the reference section; the detailed information can be found on line 813 - 815 of the revised manuscript. b) We have revised the definition of the unit for catalase activity, highlighted with a yellow background, see line - of the revised manuscript.

Comments 12: The conclusions are very weak, not sufficiently highlighting the importance of the results presented and especially the last lines (lines 776-778). Therefore, this part should be corrected in that: a) no flavor analysis is presented; b) melatonin treatments are repeated over time; c) the claim that melatonin has the greatest effect at with 180 mm irrigation should be emphasized in physiological terms but moderated considering that yield was limited (add yield data).

Response 12: Agreeing with the reviewer. We have rewritten the Conclusions chapter and highlighted it with a yellow background; detailed content can be found on line 878-903 of the revised manuscript.

Reviewer 2 Report

Comments and Suggestions for Authors

Dear Authors, I have reviewed the manuscript and have the following comments:

The Abstract is too long, almost a page. I suggest, as per MDPI's requirement, to summarize it in 200 words maximum, because this is not good. 

The lines are not numbered, although it is a formal requirement. Why?

Keywords: if it is not a proper noun and not an abbreviation, why does it start with a capital initial? I suggest that this be changed. 

The Introduction section is almost shorter than the Abstract is now. This is not right. I propose to make the Introduction chapter MUCH LONGER, especially with publication results from the last 5 years. There are 9 references in this chapter at the moment. This should be in 1 paragraph. This is too few and too many. 

In this context the whole Discussion chapter needs to be rewritten. 

Points

Specific Comments

What is the main question addressed by the research?

The effect of exogenous melatonin (MT) on the physiological responses of grapevines to stress and endogenous hormone and fruit quality under varying irrigation volumes was investigated. The results showed that MT treatment increased leaf Pro, SP and SS contents, decreased O2, H2O2 and MDA contents, and increased SOD, POD, CAT, AAO, APX, DHAR, MDHAR and GR under varying irrigation volumes, thereby promoting the accumulation of AsA, DHA and GSH and accelerating the degradation of GSSG.

Do you consider the topic original or relevant to the field? Does it address a specific gap in the field? Please also explain why this is/ is not the case.

The study of the physiological responses of grapevines is important as they are a major horticultural product in many countries.

What does it add to the subject area compared with other published material?

A genetic approach to physiological studies. The effects of osmotic adjusters, reactive oxygen species, antioxidant enzymes, antioxidant content, antioxidant enzyme activity in ascorbic acid-glutathione cycle metabolism, endogenous hormone content, related gene expression and fruit quality-related indicators were determined in grapevine leaves under different growth conditions in stages.

What specific improvements should the authors consider regarding the methodology? What further controls should be considered?

A rewrite of the abstract, the Introduction, and with it the Discussion chapter is essential. These chapters are bad and disproportionate.

Are the conclusions consistent with the evidence and arguments presented and do they address the main question posed? Please also explain why this is/is not the case.

No, since the Introduction needs to be rewritten, so does this chapter.

Are the references appropriate?

Yes, but extremely few.

Any additional comments on the tables and figures.

This is fine.

Author Response

Comments 1: The Abstract is too long, almost a page. I suggest, as per MDPI's requirement, to summarize it in 200 words maximum.

Response 1: Agreeing with the reviewer, we have rewritten the abstract and highlighted it with a yellow background; the detailed information can be found in lines 11- 24 of the revised version.

Comments 2: The abstract lines are not numbered, although it is a formal requirement. Why?

Response 2: Agreeing with the reviewer, there are line numbers on the left side of the revised manuscript.

Comments 3: Keywords: if it is not a proper noun and not an abbreviation, why does it start with a capital initial? I suggest that this be changed. 

Response 3: Agreeing with the reviewer, the error was due to a writing mistake, which has been corrected and highlighted with a yellow background; the detailed content can be found on line 25-26 of the revised version.

Comments 4: The Introduction section is almost shorter than the Abstract is now. This is not right. I propose to make the Introduction chapter MUCH LONGER.

Response 4: Agreeing with the reviewer, we have revised and supplemented the introduction section, highlighted with a yellow background; the detailed content can be found on line28- 82of the revised version.

Comments 5: In this context the whole Discussion chapter needs to be rewritten. 

Response 5: Agreeing with the reviewer. We have rewritten the Discussion chapter and highlighted it with a yellow background; detailed content can be found on line 578-761 of the revised manuscript.

Reviewer 3 Report

Comments and Suggestions for Authors

This study aims to investigate the melatonin-mediated modulation of grapevine resistance physiology, endogenous hormonal dynamics, and fruit quality under varying irrigation levels.

The introduction should provide background on the relationship between irrigation, melatonin, and the measured parameters, emphasizing the importance of including irrigation amounts.

The experimental approaches and data analyses are acceptable. The study uses a wide range of melatonin dosages and irrigation amounts, and involves many parameters, including molecular aspects, in the measurements.

The results contain several valuable elements, including comparisons of different irrigation amounts and melatonin dosages, as well as correlations between measured parameters using correlation analysis and PCA.

The discussion section is acceptable, offering suitable comparisons of the obtained results with previous literature.

In the conclusion section, the authors should highlight the future implications of their study and discuss its limitations.

The study contains interesting elements that could be worthy of publication in an international journal after appropriate revisions.

Other suggestions and comments:

Page numbering is missing, making it difficult to identify the correction locations.

Introduction

Vitis vinifera should be italicized.

"Melatonin" should be written as "melatonin" within sentences.

SOD, POD, IAA, ABA, etc., should be defined when first mentioned.

Figures

Figure 1 title: Use "melatonin on membrane permeability of grape leaves" instead of "Melatonin on Membrane Permeability of Grape Leaves." Provide full definitions for REC, pro, SP, and SS in the title.

Figures 1, 2, 3, 4, 6, 7, 8, 9, 10, 11, 12, 13, 14, 15, 16; Tables 2, 3, 4: Give explanations for W1CK, W1MT, etc., in the titles.

Figure 3 title: Include full definitions for SOD, POD, and CAT.

Figure 4 title: Include full definitions for AAO, APX, DHAR, and MDHAR.

Figure 5 title: Include full definitions for AsA, DHA, GSH, and GSSG.

References

The reference format does not follow MDPI style.

Latin names should be italicized, e.g., Coffea arabica, Arabis alpina.

There are inconsistencies in the format of authors' names.

In some instances, page numbers and/or issue numbers are missing.

Author Response

Comments 1: The introduction should provide background on the relationship between irrigation, melatonin, and the measured parameters, emphasizing the importance of including irrigation amounts.

Response 1: Agreeing with the reviewer, we have revised and supplemented the introduction section, highlighted with a yellow background; detailed content can be found on line 11-24 of the revised version.

Comments 2: In the conclusion section, the authors should highlight the future implications of their study and discuss its limitations.

Response 2: Agreeing with the reviewer, we have conducted an in-depth rewrite of the conclusion section, highlighted with a yellow background, the detailed content can be found on line 878- 903of the revised manuscript.

Comments 3: Page numbering is missing, making it difficult to identify the correction locations.

Response 3: Agreeing with the reviewer, we have added page numbers and line numbers.

Comments 4: Introduction: a) Vitis vinifera should be italicized. b) "Melatonin" should be written as "melatonin" within sentences. c) SOD, POD, IAA, ABA, etc., should be defined when first mentioned.

Response 4: Agreeing with the reviewer, we have revised the introduction section, including these three points, and highlighted them with a yellow background; the detailed content can be found on line 28- 82 of the revised version.

Comments 5: Figure 1 title: Use "melatonin on membrane permeability of grape leaves" instead of "Melatonin on Membrane Permeability of Grape Leaves." Provide full definitions for REC, pro, SP, and SS in the title.

Response 5: Agreeing with the reviewer, we have revised the title of Figure 1 and added complete definitions of the parameters. They are highlighted with a yellow background, the detailed content can be found on line110 -118 of the revised version.

Comments 6: Figures 1, 2, 3, 4, 6, 7, 8, 9, 10, 11, 12, 13, 14, 15, 16; Tables 2, 3, 4: Give explanations for W1CK, W1MT, etc., in the titles.

Response 6: Agreeing with the reviewer, we have supplemented the titles of Figures 1 (and all subsequent figures). This has been highlighted with a yellow background, for detailed changes, please refer to all the figures in the revised manuscript.

Comments 7: a) Figure 3 title: Include full definitions for SOD, POD, and CAT. b) Figure 4 title: Include full definitions for AAO, APX, DHAR, and MDHAR. c) Figure 5 title: Include full definitions for AsA, DHA, GSH, and GSSG.

Response 7: Agreeing with the reviewer, we have added complete definitions of the abbreviations in the titles of Figures 3, Figures 4, and Figures 5. These have been highlighted with a yellow background, for details, please refer to the figures in the revised manuscript.

Comments 8: References: a) The reference format does not follow MDPI style. b) Latin names should be italicized, e.g., Coffea arabica, Arabis alpina. c) There are inconsistencies in the format of authors' names. d) In some instances, page numbers and/or issue numbers are missing.

Response 8: Agreeing with the reviewer. We have revised all the content in the References section to follow the MDPI style and highlighted it with a yellow background; the detailed content can be found on line 915-1114 of the revised manuscript.

Round 2

Reviewer 1 Report

Comments and Suggestions for Authors

Although the work presents some interesting data, it deserves to be rejected because the vine is not a model plant and the study of stress response cannot be separated from the yield; otherwise it appears from Tables 3 -5 that in order to have more sugary and phenol-rich berries it is better to irrigate with 120 mm of water when the rainfall in the growing season (in a solar greenhouse ?) is 150-250 mm, in the presence of an annual evaporation between 1000 and 2000 mm (lines 768-770).

In addition, corrections are written in approximate English (e.g. ‘360mm of irrigation amount and application of water’), it is never clearly indicated whether values are expressed per fresh weight or dry weight, and enzyme activity is presented with respect to ‘Protein FW’ which means very little since enzyme activity is to be expressed with respect to the protein content in the extract analysed.

Comments on the Quality of English Language

I am not a native speaker, but with the revision, indications have been introduced that are perhaps grammatically correct but unclear in meaning, e.g. "X mm of irrigation amount and application of water" in all figure legends, or "Grape (Vitis L.)" in Abstract.

Author Response

Comments 1:  a): Although the work presents some interesting data, it deserves to be rejected because the vine is not a model plant and the study of stress response cannot be separated from the yield.  b): Otherwise it appears from Tables 3 -5 that in order to have more sugary and phenol-rich berries it is better to irrigate with 120 mm of water when the rainfall in the growing season (in a solar greenhouse?) is 150-250 mm, in the presence of an annual evaporation between 1000 and 2000 mm.

Response 1: a): We sincerely appreciate your valuable suggestions. However, in this work, we focus on fruit quality without conducting yield analysis, which is in line with our research objectives. The reasons are as follows: First of all, fruit tree production differs from field crop production. Field crops usually pursue yield, while fruit tree production is typically based on quality due to market demand. Fruit quality is the key factor that determines the economic interests of farmers, consumer choice and market value. Second, although stress response is related to yield, maintaining or improving fruit quality may be more important than increasing yield in fruit tree production. Thirdly, in grape production, due to the different climatic conditions (especially the effective accumulated temperature) in different regions, there are significant differences in the requirements for grape yield in order to ensure that the grapes can meet commercialization standards. For example, in the Dunhuang area, when the yield of grapes reaches 3000 kilograms per mu, the quality indexes such as sugar content and maturity of grapes can be ensured to meet the commercialization standards of the market. On the contrary, for regions like Zhangye, Wuwei and Lanzhou, if the yield of grapes exceeds 2000 kg per mu, it will lead to insufficient maturity, decreased sugar content and increased organic acid content, which will affect the quality of grapes and make them unable to meet the commercialization standards of the market. Therefore, the pursuit of yield cannot be generalized, and in the production of grapes, different regions usually adopt cultivation aim to limit yield, improve quality and increase efficiency, so as to ensure that grapes meet the commercial needs of the market. Our experiment is located in Huangyang Town, Wuwei City, Gansu Province. The local grape production limit is 2000 kg per mu, and the yield standard of the test site is 1500~2000 kg per mu. b): The experiment was conducted in a solar greenhouse using a delayed cultivation model. We have rechecked the total rainfall during the grape growing period (from May to October 2023), which is 97.4mm, with the following breakdown: May rainfall: 6.9mm; June rainfall: 3mm; July rainfall: 0.6mm; August rainfall: 10.4mm; September rainfall: 54.1mm; October rainfall: 22.4mm. The changes have been made, please refer to line 804-805 of the revised manuscript for details.

Comments 2:  a): In addition, corrections are written in approximate English (e.g. ‘360mm of irrigation amount and application of water’). b): it is never clearly indicated whether values are expressed per fresh weight or dry weight, and enzyme activity is presented with respect to ‘Protein FW’ which means very little since enzyme activity is to be expressed with respect to the protein content in the extract analysed.

Response 1: a): Agreeing with the reviewer, we have revised the expressions such as '360mm of irrigation amount and application of water' throughout the text. For details, please refer to the notes accompanying for all figures and tables in the revision manuscript.  b): All the values in this study are determined based on fresh weight. This information has been supplemented in the captions of Figures 1to16 and Tables 1to 4 in the revised manuscript, and highlighted with a yellow background. Please refer to the revised manuscript for specific details. In addition, the enzyme activity is not expressed relative to the fresh weight of the protein, but rather relative to the protein content in the analyzed extract. For details, please refer to Figures 3 and 4 in the revised manuscript.

Comments 3:  I am not a native speaker, but with the revision, indications have been introduced that are perhaps grammatically correct but unclear in meaning, e.g. "X mm of irrigation amount and application of water" in all figure legends, or "Grape (Vitis L.)" in Abstract.

Response 1:  Agreeing with the reviewer, we have further revised the "X mm of irrigation amount and application of water" in all figure legends to make it more understandable. For example, " (W1CK) irrigate with 360 mm of water and spray water (instead of melatonin) on the leaves". "(W1MT) irrigate with 360 mm of water and spray 150 μmol·L⁻¹ melatonin on the leaves. At the same time, we have changed "Grape (Vitis L.)" in the introduction to " Grapevine " and highlighted it with a yellow background. Please refer to line 30 of the revised manuscript for details.

Reviewer 2 Report

Comments and Suggestions for Authors

Thank you, I recommand it for publication. 

Author Response

comments: I recommand it for publication. 

response 1: Thank you.

Round 3

Reviewer 1 Report

Comments and Suggestions for Authors

Dear Authors, I am sorry but the main reason of rejection was the absence of indications concerning yield, quality cannot be separated from yield.

Author Response

评论 1: 尊敬的作者,很抱歉,被拒绝的主要原因是没有关于产量的指示,质量与产量是分不开的。

响应 1: 非常感谢您的评论。根据您的观点和学术编辑的建议,我们在讨论中解释了缺乏产量测量的合理性,并用黄色背景突出显示。有关详细信息,请参阅修订稿中的第 797-806 行。